# OXTAL: AN ALL-ATOM DIFFUSION MODEL FOR ORGANIC CRYSTAL STRUCTURE PREDICTION

**Emily Jin**[1][*][†]**, Andrei Cristian Nica**[2][*][‡]**, Mikhail Galkin**[3][§]**, Jarrid Rector-Brooks**[4,5,6]**,
Kin Long Kelvin Lee**[7][§]**, Santiago Miret**[8][§]**, Frances H. Arnold**[6]**, Michael Bronstein**[1,9]**,
Avishek Joey Bose**[1,4,11]**, Alexander Tong**[9]**, Cheng-Hao Liu**[6,10][†]

[1]University of Oxford, [2]Synteny, [3]Google, [4]Mila, [5]Université de Montréal, [6]Caltech,
[7]NVIDIA, [8]Lila, [9]AITHYRA, [10]FutureHouse, [11] Imperial College London

## ABSTRACT

Accurately predicting experimentally realizable 3D molecular crystal structures from their 2D chemical graphs is a long-standing open challenge in computational chemistry called *crystal structure prediction* (CSP). Efficiently solving this problem has implications ranging from pharmaceuticals to organic semiconductors, as crystal packing directly governs the physical and chemical properties of organic solids. In this paper, we introduce OXTAL, a large-scale 100M parameter all-atom diffusion model that directly learns the conditional joint distribution over intramolecular conformations and periodic packing. To efficiently scale OX-TAL, we abandon explicit equivariant architectures imposing inductive bias arising from crystal symmetries in favor of data augmentation strategies. We further propose a novel crystallization-inspired lattice-free training scheme, STOICHIOMET-RIC STOCHASTIC SHELL SAMPLING ($S^4$), that efficiently captures long-range interactions while sidestepping explicit lattice parametrization—thus enabling more scalable architectural choices at all-atom resolution. By leveraging a large dataset of 600K experimentally validated crystal structures (including rigid and flexible molecules, co-crystals, and solvates), OXTAL achieves orders-of-magnitude improvements over prior *ab initio* machine learning CSP methods, while remaining orders of magnitude cheaper than traditional quantum-chemical approaches. Specifically, OXTAL recovers experimental structures with conformer $\text{RMSD}_1 < 0.5$ Å and attains over 80% packing similarity rate, demonstrating its ability to model both thermodynamic and kinetic regularities of molecular crystallization.

## 1 INTRODUCTION

A landmark open challenge in computational chemistry is identifying a molecule's 3D crystal structure given knowledge of its chemical composition (Bardwell et al., 2011; Reilly et al., 2016; Hunnisett et al., 2024). In particular, given only a molecule's 2D chemical graph, *ab initio* molecular crystal structure prediction (CSP) seeks to estimate the distribution of experimentally realizable crystal packings in an accurate and scalable manner. This 3D arrangement of molecules within a periodic lattice dictates the macroscopic behavior of organic solids. For instance, in pharmaceuticals, crystal packing governs solubility, bioavailability, and the long-term stability of active ingredients (Schultheiss & Newman, 2009; Chen et al., 2011); in material science, intermolecular geometry dictates charge transport, porosity, and optical response—enabling applications across electronics, photonics, sensing, and energy storage (Zhang et al., 2018; Wang et al., 2019).

The complexity of CSP is underscored by the nature of crystal formation, wherein experimentally realized structures often occupy local minima of a highly non-smooth and well-separated Gibbs free energy landscape (Figure 3(a)). This thermodynamic energy landscape is determined by the competition between the *intramolecular* interactions that set the molecule's own (flexible) conformation

---

[*]Equal Contribution.

[†]Corresponding authors: `emily.jin@cs.ox.ac.uk`; `chl@caltech.edu`

[‡]Work completed while at Mila.

[§]Work completed while at Intel Labs.

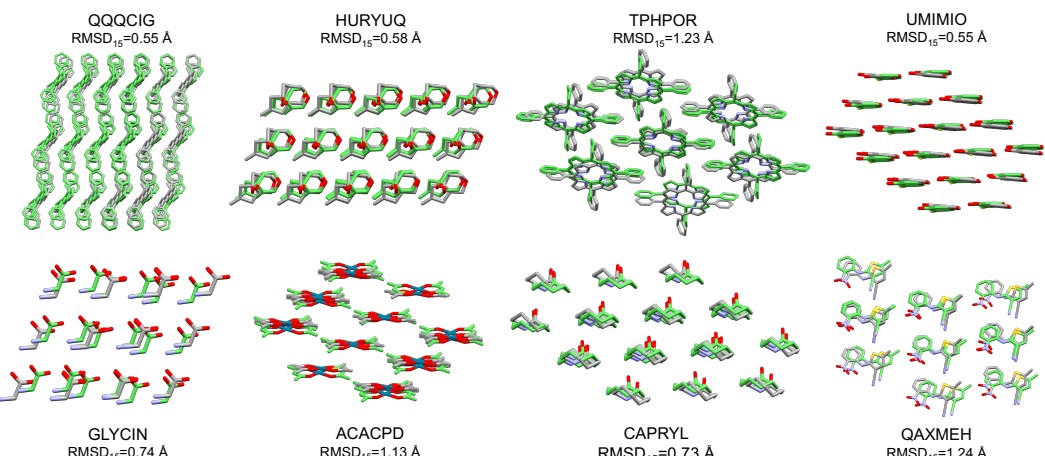

Figure 1: Molecular crystal structures generated by OXTAL (color) compared to ground truth (grey).

and the long-range and weak *intermolecular* forces that dictate how molecules pack together periodically (Chernov, 2012). As a result, classical CSP approaches combine a search procedure (e.g., enumeration or evolutionary algorithms) with oracle access to energy/ranking models, such as force fields or quantum-chemical density functional theory (DFT) (Engel & Dreizler, 2011; Hunnisett et al., 2024). However, classical CSP approaches often fail to capture realistic *kinetic* conditions that lead to the *distribution* of experimentally sampled energy minima. Consequently, these methods require the generation and optimization of ~1,000 to 100,000 structures per molecule—the majority of which struggle to go beyond unfavourable local energy minima despite extensive computation.

Recently, *generative* approaches to modeling atomic systems–e.g., AlphaFold3 (Abramson et al., 2024) for biomolecules and MatterGen (Zeni et al., 2025) for inorganic materials–have demonstrated their ability to capture intricate 3D atomistic interactions directly from data. Molecular CSP generalizes both of these, as proteins and inorganic crystals have a smaller set of interactions; proteins pack into lattices under strong intramolecular constraints mostly governed by their backbones (Branden & Tooze, 2012), and inorganic crystals possess strong covalent or ionic bonds across a smaller number of atoms (Figure 2a). Protein generative models also heavily rely on biological priors based on evolutionary information captured by multiple-sequence alignment (Jumper et al., 2021). In contrast, small-molecule crystals span chemically diverse scaffolds, exhibit rich conformational flexibility, and often contain many molecular copies within a unit cell (Figure 2b). Accurately capturing these interactions requires a large and diverse training set, highly expressive yet efficient to sample models, and training schemes that consider periodic interactions in a crystal lattice without impeding scalability for large model training.

**Main contributions**. In this paper, we introduce OXTAL, an all-atom diffusion transformer model for molecular CSP. Conditioned solely on the 2D molecular graph, OXTAL learns to sample experimentally realistic crystal structures with accurate descriptions of molecular conformers as well as their periodic packing. To enable large-scale modeling, we train OXTAL on over 600k experimental molecular crystal structures spanning rigid and flexible molecules, co-crystals, and solvates. OXTAL builds on recent advances in all-atom generative modeling (Abramson et al., 2024), discarding symmetry representations of lattice vectors and associated crystal symmetries in favor of training directly on Cartesian coordinates and employing SE(3) data augmentation. We further introduce STOICHIO-METRIC STOCHASTIC SHELL SAMPLING ($S^4$), a novel lattice-free training scheme that retains the rich information of long-range interactions. We summarize our key contributions as follows:

- We present OXTAL, the first large-scale all-atom diffusion model for molecular CSP, that samples molecular crystal packing directly from 2D molecular graphs (§3).
- We introduce a crystallization-inspired training scheme for periodic structures, $S^4$, which removes explicit lattice parametrization and enables more scalable training (§3.1).
- Empirically, OXTAL significantly outperforms existing ML-based *ab initio* CSP methods, achieving $RMSD_1 < 0.5$ Å and packing similarity rates above $80\%$ within 30 samples (§4.1), while being several orders of magnitude cheaper than traditional quantum chemical methods in DFT (§4.2).
- We provide additional chemical analysis highlighting OXTAL's ability to capture diverse intramolecular and intermolecular interactions, including crystal polymorphs, and generalize to complex co-crystal and biomolecular interactions (§4.3).

## 2 BACKGROUND AND PRELIMINARIES

### 2.1 CRYSTAL REPRESENTATIONS

Formally, a periodic crystal structure $\mathcal{C}$ is defined by a pair $(L, \mathcal{B})$. The first component $L \in \mathbb{R}^{3 \times 3}$ defines the lattice vectors forming a parallelepiped known as the unit cell. The second component $\mathcal{B} = \{(z_i, u_i)\}_{i=1}^N$ is the basis, which consists of the $N$ atoms within this unit cell. Each atom is described by its species $\{z_i\}_{i=1}^N$ and its fractional coordinates $\{u_i \in [0, 1)^3\}_{i=1}^N$ relative to the lattice vectors, or equivalently, its Cartesian coordinates $L u_i$. For molecular crystals, $\mathcal{B}$ naturally decomposes into $Z$ molecules. The connectivity of each molecule $m_k$ is given by a graph $\{g_k = (V_k, E_k)\}_{k=1}^Z$ (vertices/atoms $V_k$ and edges $E_k$; species labels $z_i$ are carried by the vertices). We next recall the symmetries present in the periodic crystal structure $\mathcal{C}$.

**Symmetries**. Given $\mathcal{C} = (L, \mathcal{B})$ where $(u_i \in \mathbb{T}^3 := \mathbb{R}^3/\mathbb{Z}^3)$, the Cartesian positions of atoms are,

$$X(L, \mathcal{B}) = \{ L(n + u_i) : n \in \mathbb{Z}^3, i = 1, \dots, N \}. \tag{1}$$

Furthermore, two descriptions $(L, \mathcal{B})$ and $(L', \mathcal{B}')$ encode the same structure if and only if there is a group action $g$ in 3D, $g := (R, t) \in \mathrm{SE}(3)$ such that, $X(L', \mathcal{B}') = g \circ X(L, \mathcal{B})$.

A periodic crystal admits an *asymmetric unit* $\mathcal{A}$, the minimal subset that recovers the entire unit cell by applying symmetry transformations of the crystal's space group. Conversely, a supercell can be obtained by an integer matrix $U \in \mathbb{Z}^{3 \times 3}$ with $m := |\det U| \geq 1$, yielding $\mathcal{C}^{(U)} = (LU, \mathcal{B}^{(U)})$, the same infinite crystal in a differing tiling. For example, $U = m I_3$ yields a cubic $m \times m \times m$ supercell. A formal summary of crystal symmetries is outlined below.

---

**Crystal representation invariances**

**(S1)** *Global translation.* For $t \in \mathbb{T}^3$, $(L, \{(z_i, u_i)\}) \mapsto (L, \{(z_i, u_i + t)\})$.

**(S2)** *Global rotation.* For $R \in \mathrm{SO}(3)$, $(L, \{(z_i, u_i)\}) \mapsto (RL, \{(z_i, u_i)\})$.

**(S3)** *Permutation (reindexing).* For $\zeta \in S_N$, $(L, \{(z_i, u_i)\}) \mapsto (L, \{(z_{\zeta(i)}, u_{\zeta(i)})\})$.

**(S4)** *Unit-cell change.* Let $U \in \mathbb{Z}^{3 \times 3}$ with $m := |\det U| \geq 1$.
- *Unimodular basis change* ($m = 1$, $U \in \mathrm{GL}(3, \mathbb{Z})$):

$$(L, \{(z_i, u_i)\}) \mapsto (LU, \{(z_i, U^{-1} u_i)\}), \quad (LU)(n + U^{-1} u_i) = L(Un + u_i).$$

- *Supercell expansion* ($m > 1$): Let $\mathcal{R}(U) \subset \mathbb{T}^3$ be a fixed set of coset representatives for $\mathbb{Z}^3/U\mathbb{Z}^3$ (so $|\mathcal{R}(U)| = m$). Then

$$(L, \{(z_i, u_i)\}) \mapsto \Big( LU, \{(z_i, U^{-1}(u_i + r)) : i = 1, \dots, N, r \in \mathcal{R}(U)\} \Big),$$

since $(LU)(n + U^{-1}(u_i + r)) = L(Un + u_i + r)$.

---

**Molecular crystallization.** Let us denote $g = \{g_k = (V_k, E_k)\}_{k=1}^Z$ as the set of molecular graph(s) and $\mathcal{X}(g)$ as the set of periodic, all-atom crystal structures compatible with $g$.

Due to the invariances, the physically distinct configurations form a quotient space $\mathcal{M}(g) = \mathcal{X}(g)/\sim$. *Ab initio* CSP can then be posed as conditional probabilistic inference over equivalence classes $[\mathcal{C}] \equiv [(L, \mathcal{B})] \in \mathcal{M}(g)$. An approximation of the experimentally realized distribution under crystallization conditions $\aleph$ is:

$$p_\aleph([\mathcal{C}] \mid g) \propto \kappa_\aleph([\mathcal{C}] \mid g) \exp(-\beta \Delta G([\mathcal{C}]))$$

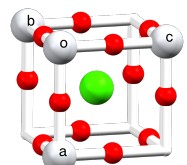 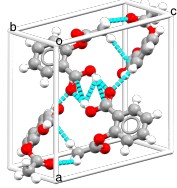

(a) CaTiO$_3$ (inorganic)    (b) Asprin (organic)

Figure 2: Molecular crystals consist of distinct molecules held together via long-range, weak interactions. They typically contain many atoms per unit cell and unknown molecule copies $Z$.

where $\Delta G$ is the Gibbs free energy (thermodynamics), $\kappa_\aleph$ summarizes kinetic accessibility (nucleation and growth pathways), $\beta = \frac{1}{k_\mathrm{B} T}$, $k_B$ is the Boltzmann constant, and $T$ is temperature.

In practice, learning and sampling must respect the symmetry of $\mathcal{M}(g)$, handle strong coupling between intramolecular conformation and intermolecular packing (especially in flexible molecules), navigate rugged kinetic and energy landscapes arising from large unit cells (often > 100 atoms) and *weak, long-range* interactions, and marginalize over unknown $Z$.

These challenges are distinct from inorganic crystals, which are characterized by continuous, strong ionic and covalent bonds, no molecular conformers, and often < 30 atoms per unit cell (Figure 2).

## 2.2 CONTINUOUS-TIME DIFFUSION MODELS

A diffusion model solves the generative modeling problem by being the solution to a (Itô) stochastic differential equation (SDE) (Øksendal, 2003),

$$\mathrm{d}\mathbf{X}_t = f_t(\mathbf{X}_t)\,\mathrm{d}t + \sigma_t \mathrm{d}\mathbf{W}_t, \quad \mathbf{X}_0 \sim p_0. \tag{2}$$

In Equation (2), we use boldface to denote random variables and normal script to denote functions or samples. The function $f_t : \mathbb{R}^d \to \mathbb{R}^d$ corresponds to the average direction of evolution, while $\sigma_t : \mathbb{R} \to \mathbb{R}$ is the diffusion coefficient for the Wiener process $\mathbf{W}_t$. By convention, this is termed the forward noising process that starts from time $t = 0$ and progressively corrupts data until the terminal time $t = 1$ where we reach a structureless prior $p_1(x_1) := \mathcal{N}(0, I)$.

Under mild regularity assumptions, forward SDEs of the form of Equation (2) admit a time reversal, which itself is another SDE in the reverse direction $t = 1$ to $t = 0$ and transmutes a prior sample $x_1 \sim p_1$ to a sample from the target distribution $x_0 \sim p_0$ (Anderson, 1982),

$$\mathrm{d}\mathbf{X}_t = \left(f_t(\mathbf{X}_t)\,\mathrm{d}t - \sigma_t^2 \nabla_x \log p_t(\mathbf{X}_t)\right)\,\mathrm{d}t + \sigma_t \mathrm{d}\overline{\mathbf{W}}_t, \quad \mathbf{X}_1 \sim p_1. \tag{3}$$

Here $\overline{\mathbf{W}}_t$ is another standard Wiener process, and the *reverse-time* SDE shares the same time marginal density $p_t$ as the forward SDE. Critically, the forward and reverse SDEs are linked via the Stein score $\nabla_x \log p_t(x_t)$, which is the key quantity of interest in the design of diffusion models. More precisely, diffusion models estimate the score function by forming an $\ell_2$-regression objective using a denoising network $D_\theta(x_t, t)$. For example, for the variance exploding (VE) SDE family of forward processes: $p_t = p_0 * \mathcal{N}(0, \sigma_t^2)$, this corresponds to learning the set of optimal denoisers, i.e., the set of conditional expectations, across time $D(x_t, t) = \mathbb{E}\left[\mathbf{X}_0 | \mathbf{X}_t = x_t\right], \forall t \in [0, 1]$. Owing to the nature of Gaussian convolution, we can convert this to a simple *simulation-free* training *conditional* objective for any Bregman divergence with convex F, $\mathbb{B}_\mathrm{F}$ (Holderrieth et al., 2025, Prop. 2)

$$\mathcal{L}_c(\theta) = \mathbb{E}_{t \sim \mathcal{U}(0,1), x_0 \sim p(x_0), x_t \sim p_t(x_t | x_0)}\left[\lambda(t)\mathbb{B}_\mathrm{F}(x_0, D_\theta(x_t, t))\right], \tag{4}$$

where $\lambda(t) : [0, 1] \to \mathbb{R}_+$ is any weighting function. For a sufficiently expressive family of denoisers $\mathcal{H} = \{D_\theta : \theta \in \Theta\}$, the minimizer to Equation (4) is $D_\theta^*(x_t, t) = D(x_t, t) = \mathbb{E}\left[\mathbf{X}_0 | \mathbf{X}_t = x_t\right]$. Given a trained denoising model, diffusion models generate samples at inference by simulating the reverse-time dynamics of (3) with the learned score $s_\theta$ computed as $s_\theta \leq (D_\theta(x_t, t) - x_t)/\sigma_t^2$.

## 3 OXTAL

We now describe OXTAL, our all-atom diffusion model for molecular CSP. As outlined in §2.1, existing inorganic CSP models (Miller et al., 2024; Gasteiger et al., 2021) that rely on equivariant architectures and explicit unit-cell parametrizations face scalability challenges for large molecular crystals with unknown multiplicity $Z$. We thus introduce $S^4$ training, which exposes the model to long-range periodic cues without ever parametrizing a lattice (§3.1). Second, we implement a high-capacity, non-equivariant Transformer with data augmentation and strong molecular embeddings to capture symmetries (§3.2). Together, these choices decouple what to generate (conformations and packing) from how the crystal is represented during training (unit cell, $Z$, etc.).

## 3.1 STOICHIOMETRIC STOCHASTIC SHELL SAMPLING ($S^4$)

Crystallization is a local-to-global process: once molecules approach contact distances, weak but specific interactions induce recurring motifs that propagate periodically. Learning to denoise such local consistent neighborhoods should therefore recover larger periodicity at inference time. Training on such subsampled blocks reduces token size, provides natural augmentation, and mirrors the partial observability of nucleation and growth (Figure 3(b)).

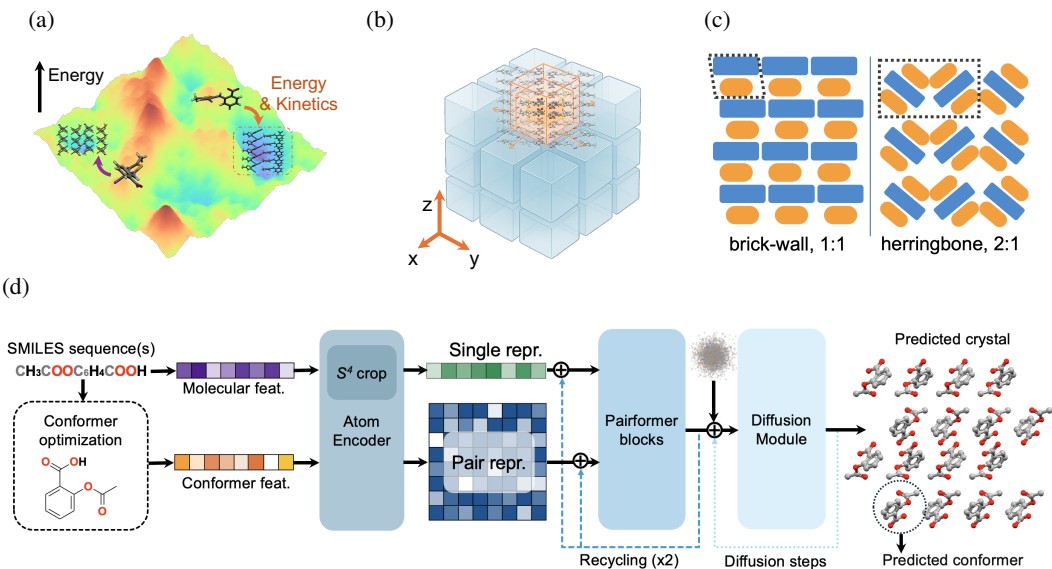

Figure 3: (a) Schematic of a rugged crystallization Gibbs free energy landscape with many local minima. Kinetic conditions often dictate which experimental minimum is formed. (b) Molecular crystallization, showing *nucleation* and *growth* in successive layers, which is the inspiration for $S^4$. (c) Common packing motifs exemplified in co-crystal polymorphs with 1:1 and 2:1 stoichiometric ratio. (d) Overview of OXTAL architecture.

We formalize this idea in $S^4$. Let $\mathcal{C}^{(U)} = (LU, \mathcal{B}^{(U)})$ be a supercell with molecules $m \in \mathcal{M}$, where $X(m) \subset \mathbb{R}^3$ denotes $m$'s non-hydrogen Cartesian coordinates. We next define the *minimum-image* intermolecular distance between two molecules, $d_{\min}(m, m') = \min_{x \in X(m), x' \in X(m')} \|x - x'\|_2$.

Given a fixed contact radius $r_{\mathrm{cut}}$, $S^4$ builds concentric shells $\mathcal{S}$ around a uniformly-sampled central molecule $m_c \sim \mathrm{Uniform}(\mathcal{A})$ based on the molecular contact graph induced by $d_{\min}(\cdot, \cdot) \le r_{\mathrm{cut}}$:

$$\mathcal{S}_k(m_c) = \{m_i \in \mathcal{M} : kr_{\mathrm{cut}} \le d_{\min}(m_c, m_i) < (k+1)r_{\mathrm{cut}}\}, \quad k = 0, 1, 2, \ldots \tag{5}$$

We sample the number of shells $K \sim \mathrm{Uniform}(0, k_{\max})$ with probability $(1 - p_{\max})$, and define the block of molecules $V_K = \cup_{j=0}^K S_j$. We cap block size by a token budget $T_{\max}$ (atoms). If $|V_K| \le T_{\max}$ we accept $V_K$; otherwise we choose the smallest $K^\star$ with $|V_{K^\star+1}| > T_{\max}$, and if no full shells fit, we subsample the frontier $S_{K^\star}$ to meet the budget while preserving the crystal's molecular stoichiometry (Figure 3(c)). For example, if molecule type $i$ appears $N_i^{\mathrm{ASU}}$ times in $\mathcal{A}$ and $N_i$ times in $S_{K^\star}$, we sample molecules of type $i$ in $S_{K^\star}$ with weight $\omega_i \propto (N_i^{\mathrm{ASU}}/N_i)$. The resulting set is our training crop, denoted $\mathbf{A}_{\mathrm{crop}}$.

Compared to centroid- or $k$NN-based heuristics, this "shell cropping" better respects local interaction networks by respecting anisotropic packing motifs and interactions beyond the strongest ones (Figure 9), while mitigating truncation biases common in large-graph learning (Zeng et al., 2020). §D.1 shows that training with $S^4$ outperforms $k$NN and centroid cropping methods, and §F.1 shows that training with $S^4$ can generalize to long-range periodicity beyond training token sizes. Hyperparameter ablations of $r_{\mathrm{cut}}$ can be found in §D.2. Finally, we bound the error due to cropping. In the next proposition (see §A for proof), we show that the error in the loss due to cropping with $S^4$ decreases with the cube root of the number of tokens.

**Proposition 1.** *Let* $\partial \mathbf{A}_{crop} = \{\{u, v\} \in E : u \in \mathbf{A}_{crop}, v \notin \mathbf{A}_{crop}\}$ *represent the boundary of* $\mathbf{A}_{crop}$. *Denote the number of atoms in a volume $C$ as $T(C)$. Let* $L_\partial(\mathbf{A}_{crop}) = \sum_{\{u,v\} \in \partial \mathbf{A}_{crop}} \mathcal{L}(u, v)$ *be the boundary loss. Assuming* $\exists r^0$ *s.t.* $\mathcal{L}(u, v) = 0, \forall u, v$ *s.t.* $\|u - v\| > r^0$, *i.e. $\mathcal{L}$ is local, and there exist $0 < a \le b < \infty$ s.t. $a|S| \le T(S) \le b|S|$ for any $S \subseteq V$. Then,*

$$\frac{L_\partial(\mathbf{A}_{crop})}{T(\mathbf{A}_{crop})} = O((1 + r_{cut})T(\mathbf{A}_{crop})^{-1/3}) \tag{6}$$

## 3.2 MODEL ARCHITECTURE

OXTAL is comprised of: (1) an **Atom encoder** that embeds both physical and structural information; (2) a **Pairformer trunk** that propagates information across all atoms in the crop; (3) a **Diffusion module** that takes in the single and pairwise representations and outputs a generated crystal structure. The overall architecture of OXTAL is depicted in Figure 3(d).

**Atom encoder**. Given an input SMILES sequence $s$, we generate a 3D conformer with RDKit ETKDG followed by relaxation by the semi-empirical quantum chemical method GFN2-xTB (Pracht et al., 2020). The atomic number, positions, formal charges, Mulliken partial charges, and bond information are used as *feature embeddings* for the model. OXTAL is not particularly sensitive to the feature conditioning conformer coordinates (§E). Finally, we resolve ambiguity of identical molecular copies via relative position encoding on entity identifiers (Abramson et al., 2024).

**Pairformer trunk**. We adapt existing state-of-the-art architectures for protein folding to generate single and pair representations for each molecular crystal (Abramson et al., 2024). Instead of tokenizing protein residues, we simplify the tokenization such that each token directly represents a single atom $a_i$ in the molecule. We then apply the Pairformer stack from AlphaFold3, which leverages triangular self-attention to update the single and pair representations. Unlike AlphaFold2, which relied on the equivariant Evoformer (Jumper et al., 2021) architecture, this simpler Pairformer module is not explicitly equivariant, allowing for training on larger sequences.

**Diffusion module**. The design of our diffusion module follows that of AlphaFold3 (Abramson et al., 2024), consisting of an atom attention encoder which combines token information given by the pairformer with an encoded representation of $x_t$. This is followed by a large 70M parameter diffusion transformer (Peebles & Xie, 2023), before a final atom attention decoder predicts the denoised atomic positions. We broadly follow Karras et al. (2022) for pre-conditioning of model inputs. A diffusion module size ablation is further provided in §D.3.

## 3.3 TRAINING

OXTAL is trained using a procedure similar to those successfully employed for protein structure prediction (Abramson et al., 2024). The training objective is a composite loss designed to capture both the global structure and the accuracy of the local chemical environment. This loss is comprised of two main components: (1) a mean squared error loss $\mathcal{L}_{\text{mse}}$, (2) a smooth local difference distance test $\mathcal{L}_{\text{sLDDT}}$ as defined in Abramson et al. (2024). Both losses compare the predicted structure $\hat{x}_0 := D_\theta(x_t, t)$ and an aligned ground truth structure $x_0^{\text{align}} = \text{align}(x_0, \hat{x}_0)$, and the latter emphasizes the pairwise interactions within the crop via a surrogate of the interatomic distances. To round off our training, we also include a distogram loss on a separate head branching from the trunk $\mathcal{L}_{\text{dist}}(\hat{d}, d)$ to ensure the trunk output contains binned pairwise distance information. Additional details for computing component losses are provided in §B.2.3. The final loss is then a weighted sum of these components:

$$\mathcal{L}(\theta) = \mathbb{E}_{t \sim \mathcal{U}(0,1), x_t \sim p_t(x_t | x_0^{\text{align}})} \left[ \mathcal{L}_{\text{mse}}(\hat{x}_0, x_0^{\text{align}}) + \mathcal{L}_{\text{sLDDT}}(\hat{x}_0, x_0^{\text{align}}) \right] + \lambda_{\text{dist}} \mathcal{L}_{\text{dist}}(\hat{d}, d). \quad (7)$$

We next curate a training dataset from the Cambridge Structural Database (CSD) that contains $\sim 600k$ crystals. Specific details regarding model training and configuration are outlined in §B.

## 4 EXPERIMENTS

We evaluate OXTAL on several different datasets for molecular CSP, comparing against a range of ML-based (§4.1) and DFT energy-based methods (§4.2). These results are complemented with a broader chemical survey (§4.3). See §C for exact specifications.

**Baselines**. We compare against *ab initio* ML methods and energy-based methods. For ML methods, most models for inorganic CSP are incompatible, so we evaluate against the pre-trained AssembleFlow, a molecular CSP method that infers crystal packing from rigid-body molecules (Guo et al., 2025) and A-Transformer, an all-atom transformer flow matching model (§C.3.1). Additional results for zero-shot inference from AlphaFold3 are presented in §C.3.3. For energy-based methods, we compare against computational chemistry baselines that submitted to CCDC's 5th, 6th, and 7th CSP blind tests (Bardwell et al., 2011; Reilly et al., 2016; Hunnisett et al., 2024).

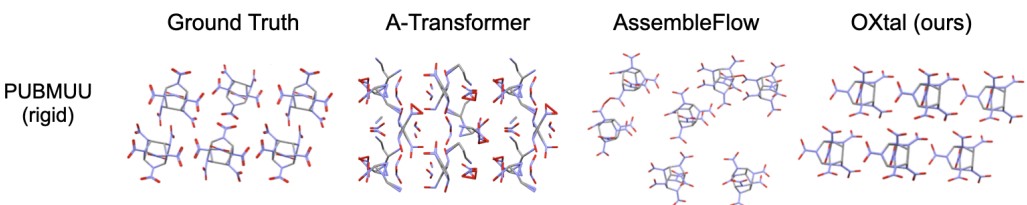

Figure 4: Example crystal packing generated by A-Transformer, AssembleFlow, and OXTAL.

Table 1: Performance of *ab initio* ML models on rigid and flexible molecular CSP in 30 samples. OXTAL achieves an order of magnitude improvement and is the only model able to approximately solve any crystals in the flexible dataset.

| Model | $Col_S \downarrow$ | $Pac_S \uparrow$ | $Pac_C \uparrow$ | $Rec_S \uparrow$ | $Rec_C \uparrow$ | $\widetilde{Sol}_C \uparrow$ |
|---|---|---|---|---|---|---|
| | | | Rigid Dataset | | | |
| A-Transformer | 0.731 | 0.015 | 0.060 | 0.033 | 0.120 | 0.060 |
| AssembleFlow | 0.524 | 0.001 | 0.040 | 0.211 | 0.760 | 0 |
| **OXTAL** | **0.011** | **0.873** | **1.000** | **0.737** | **0.960** | **0.300** |
| | | | Flexible Dataset | | | |
| A-Transformer | 0.900 | 0.001 | 0.020 | 0 | 0 | 0 |
| AssembleFlow | 0.883 | 0 | 0 | 0.021 | 0.140 | 0 |
| **OXTAL** | **0.097** | **0.291** | **0.900** | **0.048** | **0.400** | **0.220** |

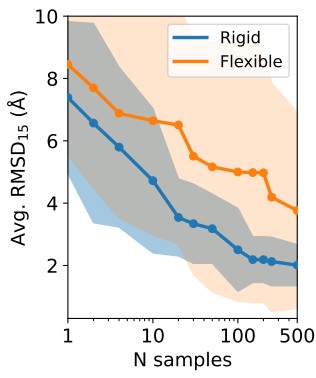

Figure 5: OXTAL sample efficiency for 10 rigid & flexible molecules.

**Metrics**. We adopt standard CSP metrics and report both *sample-* and *crystal-level* scores:

1. *Collision rate* ($Col_S$): Fraction of generated samples with any intermolecular distance $< r_w - 0.7\,\text{Å}$ where $r_w$ is the sum of atomic van der Waals radii (Cordero et al., 2008). Lower is better.
2. *Packing similarity rate* ($Pac_S$ and $Pac_C$): Using CSD COMPACK, a sample's packing is *partially* similar if at least 8 of 15 molecules *could* be aligned to an experimental cluster (cluster size per CSP5; see §C). $Pac_C$ is the fraction of targets with at least one match.
3. *Conformer recovery* ($Rec_S$ and $Rec_C$): $RMSD_1 < 0.5\,\text{Å}$ (non-hydrogen) to a solid-state conformer. $Rec_S$ averages over all samples; $Rec_C$ is the fraction of targets with at least one match.
4. *Approximately solved* ($\widetilde{Sol}_C$): *Any* collision-free, packing-similar sample with $RMSD_{15} < 2\,\text{Å}$ on a 15-molecule cluster.

## 4.1 CSP FOR RIGID AND FLEXIBLE MOLECULES

First, we compare OXTAL to *ab initio* ML models on two different test sets comprised of representative rigid and flexible molecules with ground-truth structures in CSD (see §C.2 for details). For each crystal target, we generate $n_S = 30$ samples using each model. Training-time exclusions ensure no target appears in the training sets of OXTAL or A-Transformer; AssembleFlow uses a public checkpoint that may include overlap. DFT baselines are omitted here due to prohibitive cost.

**Results**. Table 1 shows OXTAL outperforms existing ML methods across all metrics on both datasets. Intramolecularly ($Rec_C$), OXTAL recovers up to 90% of solid-state molecular conformers; intermolecularly, $Col_S$ is near zero on rigid targets and low on flexible ones. OXTAL's predicted packings also attain strong packing similarity rate against experimental structures, with approximate solves for both rigid and flexible molecules. Qualitatively (Figure 4), A-Transformer struggles to capture meaningful conformers despite being given $Z$, highlighting the limitations of a unit cell based model. AssembleFlow generates molecular assemblies with large spatial separations that lack periodicity (also reflected in low $Pac_C$ and $Pac_S$ scores), along with frequent interatomic clashes.

**Sample efficiency**. For downstream screening and design, few-sample success is critical. OXTAL exhibits a log-linear improvement in $RMSD_{15}$ among packing-similar predictions as $n$ increases (Figure 5), with several rigid targets reaching $Pac_C$ and $\widetilde{Sol}_C$ with $n < 10$. This suggests the sampler (i) often lands near the correct motif and (ii) refines global periodicity with additional draws.

## 4.2 CCDC CSP BLIND TESTS

Every few years, CCDC holds a CSP blind test competition, which invites leading computational chemistry groups to solve a handful of hidden crystal structures (Bardwell et al., 2011; Reilly et al., 2016; Hunnisett et al., 2024). We therefore evaluate OXTAL on structures from the three most recent (5th, 6th, and 7th) blind tests. Metrics and experiment set up follow §4.1. We compare OXTAL and other *ab initio* ML baselines against the aggregate of reported expensive DFT-based submissions ($DFT_{avg}$). See §C.4 for additional details and §I for per-structure DFT results.

**Results.** Table 2 shows that OXTAL strongly outperforms *ab initio* ML baselines, and achieves the best or second best performance across all three tests using only 30 samples per target. While DFT methods attain higher approximate solve rates, when OXTAL is allowed to generate the same number of samples as the DFT methods for CSP Blind Test 5, it matches the approximate solve rate of $DFT_{avg}$ (§F.2). OXTAL consistently scores the best in terms of packing similarity rate. From a per-sample basis, only $6-30\%$ of DFT samples match the packing cluster, whereas OXTAL generally recapitulates the packing structure $(48-67\%)$. This suggests that while DFT identifies many local energetic minima, OXTAL can better capture the joint energy and kinetic features that determine which motifs are more likely to be formed. Of the hundreds of submitted predictions (and potentially many more generated but unsubmitted structures), only a small percentage of DFT-identified minima are close to ground truth structures (Figure 6). Predicting the correct experimental structure in few shots is crucial for downstream discovery applications, and OXTAL reliably approximates crystal packings *sans* any ranking methods.

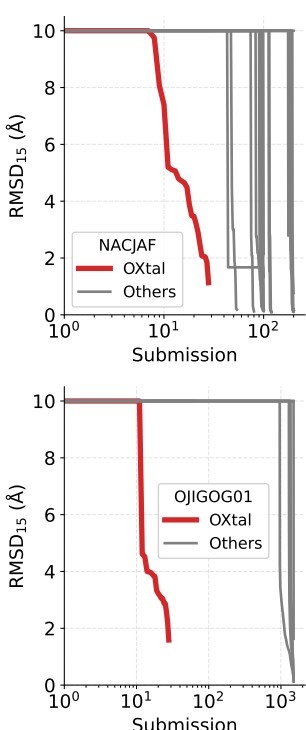

Figure 6: OXTAL, for targets shown, achieves similar $RMSD_{15}$ with fewer submissions compared to DFT.

**Inference cost.** A major pitfall of DFT-based methods is the extremely high computational cost required to run the atomistic simulations. Recently, CCDC raised concerns over the 46 million CPU core hours that were reportedly utilized by submitted methods in CSP7 to solve only 8 crystal targets

Table 2: Results for the 5th, 6th, and 7th CCDC CSP blind tests. Classical chemistry methods are aggregated as $DFT_{avg}$. The best model is **bolded**, and the second best is underlined.

| Model | $n_S$ | $Col_S \downarrow$ | $Pac_S \uparrow$ | $Pac_C \uparrow$ | $Rec_S \uparrow$ | $Rec_C \uparrow$ | $\widetilde{Sol_C} \uparrow$ |
|---|---|---|---|---|---|---|---|
| | | | CSP Blind Test 5 | | | | |
| A-Transformer | 30 | 0.833 | 0 | 0 | 0 | 0 | 0 |
| AssembleFlow | 30 | 0.717 | 0 | 0 | 0.150 | 0.500 | 0 |
| $DFT_{avg}$ | 464 | 0.039 | 0.323 | 0.661 | **0.784** | 0.733 | **0.544** |
| **OXTAL** | 30 | **0.006** | **0.667** | **0.833** | 0.572 | **0.833** | 0.167 |
| | | | CSP Blind Test 6 | | | | |
| A-Transformer | 30 | 0.967 | 0 | 0 | 0 | 0 | 0 |
| AssembleFlow | 30 | 0.800 | 0 | 0 | 0.073 | 0.200 | 0 |
| $DFT_{avg}$ | 83 | 0.067 | 0.183 | 0.520 | **0.655** | 0.560 | **0.496** |
| **OXTAL** | 30 | **0.013** | **0.660** | **1.000** | 0.160 | **0.600** | 0.200 |
| | | | CSP Blind Test 7 | | | | |
| A-Transformer | 30 | 0.950 | 0 | 0 | 0 | 0 | 0 |
| AssembleFlow | 30 | 0.808 | 0 | 0 | 0.063 | 0.250 | 0 |
| $DFT_{avg}$ | 868 | 0.072 | 0.058 | 0.511 | **0.337** | **0.449** | **0.421** |
| **OXTAL** | 30 | **0.021** | **0.483** | **0.875** | 0.129 | 0.375 | 0.125 |

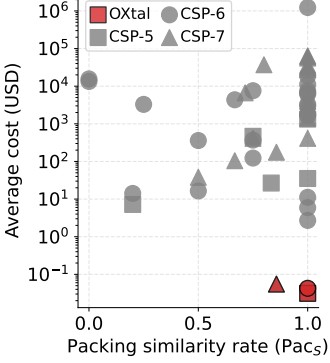

Figure 7: Packing similarity rate per crystal attempted relative to average inference cost (in $USD) for submitted CCDC competition methods. OXTAL is denoted in red. Costs are normalized to a single on-demand AWS instance from Sept. 2025 (see §C.4.1).

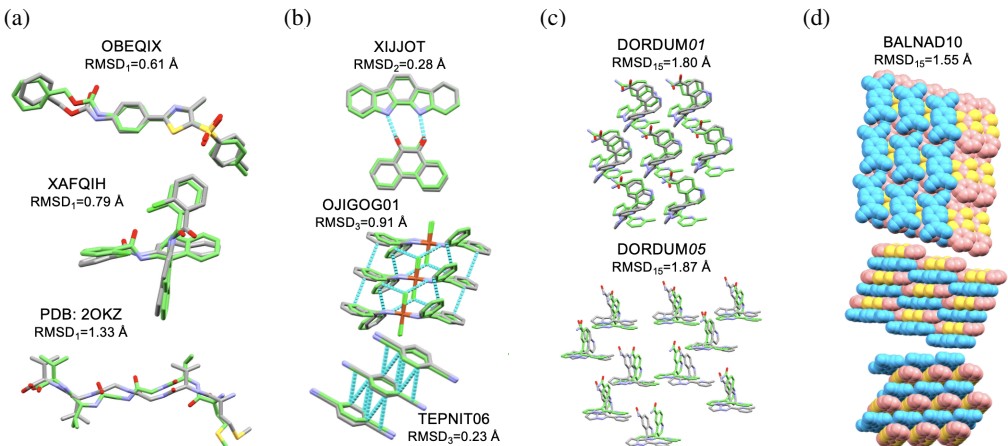

Figure 8: OXTAL (green) captures experimental (grey) (a) intramolecular and (b) intermolecular interactions in drug-like molecules, peptides, semiconductors, and catalysts. OXTAL can further infer (c) distinct experimental polymorphs as well as (d) co-crystals.

(Hunnisett et al., 2024). Unlike traditional DFT methods that require new simulations for each new molecule, OXTAL's upfront training cost (§B) is amortized at inference, enabling efficiently sampling for new molecules. Using standardized on-demand cloud pricing (§C.4.1), OXTAL *is over an order of magnitude cheaper at inference time*, while maintaining strong packing similarity rates (Figure 7). Given CCDC's call for more efficient CSP algorithms, OXTAL's cost profile enables broad screening and dense posterior sampling before optional physics-based refinement.

## 4.3 SURVEY OF CHEMICAL INTERPRETABILITY

Beyond benchmark metrics, we examine OXTAL's ability to reproduce chemically meaningful intramolecular and intermolecular features in practically relevant crystals. Figures 1 and 8 and §H highlight accurate packings ($\mathrm{RMSD}_{15} < 1.5\,\text{Å}$) across diverse rigid and flexible chemotypes: drug-like molecules (HURYUQ), polymer precursors (CAPRYL), organometallics (ACACPD), $\pi$-conjugated materials (QQQCIG), and ROY (QAXMEH), the molecule which has the most known polymorphs.

**Molecular interactions**. For intramolecular interactions, OXTAL recovers *solid-state* geometries for highly flexible molecules (which are highly influenced by packing), including small-molecule drugs as well as biomolecular fragments in the Protein DataBank (e.g. $\mathrm{RMSD}_1 = 1.3\,\text{Å}$ for a 6-mer peptide with 17 rotatable bonds (2OKZ) in Figure 8(a)). For intermolecular interactions, OXTAL accurately captures both strong and weak interactions, and both in registry and lengths. Examples in Figure 8(b) include complementary hydrogen bonds in a semiconducting crystal (XIJJOT), weak Cl-H halogen bonds in an organometallic catalyst (OJIGOG), $\pi$-$\pi$ stacking in $\pi$-functional molecule (TEPNIT), and multiple weak contacts in a cluster of the flexible drug aripiprazole (MELFIT) (Figure 18(b)). Zooming out, while OXTAL may prefer planar motifs, it can still reproduce diverse packing motifs, including 1D columnar structures (e.g. BALNAD in Figure 8(d)), quasi-2D herringbone structures (e.g. ANTCEN in Figure 16), and 2D brickwork structures (e.g. UMIMIO in Figure 1).

**Polymorphs**. For molecules with many known polymorphs (including highly rotatable scaffolds), independent OXTAL samples predict distinct experimental polymorphs (e.g., the drugs galunisertib DORDUM in Figure 8(c) and indomethacin INDMET in Figure 18(c)). This suggests the sampler can *explore multiple kinetic and thermodynamic basins* rather than collapsing to a single motif. The diversity of sampled polymorphs is shown in Figure 20.

**Multi-component systems**. OXTAL is not restricted to single-component systems. OXTAL can correctly predict the interactions between electron donor and acceptors (Figures 8 and 17), which dictate these crystals' electronic properties. For the charge-transfer semiconducting co-crystals BALNAD and PERTCQ, OXTAL correctly reproduces the 1D $\pi$-stacked columns with alternating donors and acceptors, the inter-columnar brick-wall registry, as well as the $\pi$-$\pi$ stacking distances.

**Energy analysis**. Finally, we probe the energetic plausibility of OXTAL samples using single-point GFN2-xTB calculations (§H.1). Compared to the physics-based submissions to the CSD blind tests, OXTAL samples occupy a similar, relatively tight energy basin. This indicates that even without any geometry relaxations, OXTAL learns to avoid unphysical motifs and respects the coarse thermodynamic landscapes.

## 5 RELATED WORKS

**Physical approaches to crystal structure prediction**. Classical crystal structure prediction (CSP) mostly applied search-based algorithms to sample a pre-defined search space (Hunnisett et al., 2024). While many such methods have shown varied degrees of success for different applications (van Eijck, 2002; Pickard & Needs, 2011; Case et al., 2016; Tom et al., 2020; Banerjee et al., 2021), they fundamentally rely on many calls to an expensive evaluation function (e.g., energy computation using DFT) and struggle to leverage prior data effectively. Recent search and optimization approaches replace DFT with machine learning interatomic potentials (Batatia et al., 2025; Wood et al., 2025; Gharakhanyan et al., 2025). In contrast, OXTAL does not require explicit energy function calls and brings orders of magnitude improvements in speed and inference costs.

**Generative models for inorganic crystal structure generation**. Generative models have been applied for unconditional de novo generation of inorganic periodic crystals (Xie et al., 2022) and later used for generation conditioned on crystal composition (Jiao et al., 2023; 2024; Yang et al., 2024; Miller et al., 2024; Levy et al., 2025). In contrast to molecular crystals, inorganic crystals typically comprise smaller unit cells governed by strong inter-atomic bonding, resulting in rigid continuous lattices rather than flexible molecular packing.

**Protein structure prediction**. In computational structural biology, the analogous task of protein structure prediction (PSP) conditioned on a protein sequence has seen transformative progress with landmark models like AlphaFold (Jumper et al., 2021; Abramson et al., 2024) and ESMFold (Lin et al., 2023), followed by de novo protein design approaches (Watson et al., 2023; Yim et al., 2023; Bose et al., 2024). These models work with a few dozen residue types, compared to a larger chemical space in general molecular CSP, and use evolutionary information not present in molecular crystals. A more detailed discussion of related work can be found in §G.

## 6 DISCUSSION

In this paper, we introduce OXTAL, a large-scale all-atom diffusion model for 3D molecular CSP that learns the joint distribution of molecular conformations and periodic packing conditioned on 2D graphs. Discarding explicit equivariance and unit-cell parametrization for symmetry-aware augmentation and $S^4$, OXTAL learns periodic motifs from locally consistent neighborhoods at scale, enabling efficient sampling at all-atom resolution. Empirically, OXTAL achieves state-of-the-art results among *ab initio* ML methods, and attains competitive packing similarity at several orders of magnitude lower cost compared to DFT-based methods. Our chemical survey supports OXTAL's ability to capture diverse intra- and intermolecular interactions while avoiding unphysical motifs. Nevertheless, many improvements still remain. These include integrating reliable ranking, incorporation of local relaxation, conditioning on crystallization context (i.e. solvent, temperature), and further improving solve rate and sample efficiency.

## CONTRIBUTIONS STATEMENT

CL and AN conceived the idea for this project. CL, EJ, and AN led the project development. An earlier version of the model was developed by AN, CL, and MG. The current model was developed by EJ, CL, and JRB. The theoretical work was led by AT and AN, with help from EJ, ABJ, and CL. Data processing was led by CL. Baseline experiments were led by AN, CL, and EJ. Inference and chemical/energy analysis were led by CL, KL, and EJ. CL, AJB, EJ, and AT led the writing with help from all authors. CL, AT, and AJB supervised the project with input from MB, SM, and FA.

## ACKNOWLEDGMENTS

The authors thank fruitful discussions with Tara Akhound-Sadegh, Xiang Fu, Francesca-Zhoufan Li, Dinghuai Zhang, and Hatem Titi. EJ is partially funded by AstraZeneca and the UKRI Engineering and Physical Sciences Research Council (EPSRC) with grant code EP/S024093/1. AJB acknowledges partial funding through an NSERC PDF scholarship. This research is partially supported by EPSRC Turing AI World-Leading Research Fellowship No. EP/X040062/1 and EPSRC AI Hub on Mathematical Foundations of Intelligence: An "Erlangen Programme" for AI No. EP/Y028872/1. AJB is partially supported by an NSERC PDF. The authors acknowledge funding from UNIQUE, CIFAR, Amgen, NSERC, Intel, and Samsung. The research was enabled in part by computational resources provided by the Digital Research Alliance of Canada (https://alliancecan.ca), Mila (https://mila.quebec), and NVIDIA.

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

APPENDIX

## A  THEORY

**Proposition 1.** *Let $\partial\mathbf{A}_{crop} = \{\{u, v\} \in E : u \in \mathbf{A}_{crop}, v \notin \mathbf{A}_{crop}\}$ represent the boundary of $\mathbf{A}_{crop}$. Denote the number of atoms in a volume $C$ as $T(C)$. Let $L_\partial(\mathbf{A}_{crop}) = \sum_{\{u,v\}\in\partial\mathbf{A}_{crop}} \mathcal{L}(u, v)$ be the boundary loss. Assuming $\exists r^0$ s.t. $\mathcal{L}(u, v) = 0, \forall u, v$ s.t. $\|u - v\| > r^0$, i.e. $\mathcal{L}$ is local, and there exist $0 < a \le b < \infty$ s.t. $a|S| \le T(S) \le b|S|$ for any $S \subseteq V$. Then,*

$$\frac{L_\partial(\mathbf{A}_{crop})}{T(\mathbf{A}_{crop})} = O((1 + r_{cut})T(\mathbf{A}_{crop})^{-1/3}) \tag{6}$$

To prove this proposition, we first label our assumptions.

A1 **Locality.** $\exists r_0$ s.t. $L(u, v) = 0$ for all $u, v$ s.t. $\|u - v\| > r_0$

A2 **Uniform Density.** there exist $0 < a \le b < \infty$ s.t. $a|S| \le T(S) \le b|S|$ for any $S \subseteq V$.

Next we investigate the $S^4$ algorithm. We first recall the definition of shells and $\mathbf{A}_{crop}$:

$$\mathcal{S}_k(g_c) = \{g_i \in \mathcal{C}^{(U)} : kr_{cut} \le d(g_c, g_i) < (k + 1)r_{cut}\}, \quad k = 0, 1, 2, \dots \tag{8}$$

where $d$ is the distance between molecules $g_c$ and $g_i$ in the infinitely repeating lattice, and $r_{cut}$ is some predefined contact radius. Molecules from each subsequent full shell $k$ are added until $|V_{k+1}| > T_{max}$, where $T_{max}$ is the crop budget. If at least one full shell fits, we train on the crop $\mathbf{A}_{crop} = \{g_c\} \cup \bigcup_{k\in\mathcal{K}} \mathcal{S}_k$. This implies that $\mathbf{A}_{crop}$ can be well approximated by a ball centered around $g_c$ of radius $(k + 1)r_{cut}$.

We define the ball of radius $r_k = r_{cut}k$ as $\mathcal{B}_k = \bigcup_{k\in\mathcal{K}} \mathcal{S}_k$. First, we present a short lemma to show the number of neighbors of a node is bounded.

**Lemma 1.** *We define the **edge neighborhood** of $g$ as $\mathcal{N}(g) = \{\{u, v\}$ s.t. $u = g \cap \mathcal{L}(u, v) \ge 0\}$. Assuming [A1] and [A2], we have a uniform bound on the degree*

$$|\mathcal{N}(g)| \le C \tag{9}$$

*for some $C$ independent of $g$.*

*Proof.* This follows from considering a ball of radius $r_k$ around $g$. That ball has volume $V = \frac{4}{3}r_k^3$. Using [A2], we have that the token count $T(\mathcal{B}_k(g)) \le \frac{4b}{3}r_k^3$, and therefore $|\mathcal{N}(g)| < \frac{4b}{3}r_k^3$ which is independent of $g$. $\square$

We next note that for a regular lattice structure we have the following inequality for the size of the boundary relative to the total volume. Specifically for a lattice we have:

**Lemma 2.** *For $0 < a \le b < \infty$ on a 3D lattice, we have the following relationship between a sphere's surface area and volume: $a|\mathcal{B}_k|^{2/3} \le |\partial\mathcal{B}_k| \le b|\mathcal{B}_k|^{2/3}$.*

We note that the constants are due to discretization error and locality. As we approach the continuous limit (i.e. $k \to \infty$), the bounds become tight.

Next, we have to bound how well $\mathbf{A}_{crop}$ is approximated by a ball of radius $k$.

**Lemma 3.** *For some constant $0 < c < \infty$, $\partial\mathbf{A}_{crop} \le \partial\mathcal{B}_k + cr_{cut}|\mathcal{B}_k|^{2/3}$.*

*Proof.* We first note that $\mathcal{B}_k \subseteq \mathbf{A}_{crop}$ by construction of $\mathbf{A}_{crop}$. This means that we can consider $\mathbf{A}_{crop}$ as all the molecules in $\mathcal{B}_k$ plus (possibly) some additional molecules in $\mathcal{S}_k(g_c)$. We can then bound the total surface area as

$$|\partial\mathbf{A}_{crop}| \le |\partial\mathcal{B}_k| + |\partial\mathcal{S}_k(g_c)| \tag{10}$$

however we know that the surface area of molecules in $\mathcal{S}_k(g_c)$ is bounded by the number of nodes in $\mathcal{S}_k$ and some constant.

$$|\partial \mathcal{S}_k(g_c)| \leq c|\mathcal{S}_k| \tag{11}$$

$$= c((r_{\text{cut}}(k+1))^3 - (r_{\text{cut}}k)^3) \tag{12}$$

$$= cr_{\text{cut}}^3(3k^2 + 3k + 1) \tag{13}$$

$$\leq cr_{\text{cut}}^3 k^2 \tag{14}$$

$$\leq cr_{\text{cut}}|\mathcal{B}_k|^{2/3} \tag{15}$$

which combined with equation 10 proves the lemma. $\square$

We are now ready to prove the main proposition.

*Proof.* Assuming $\mathcal{L}_\partial(\mathbf{A}_{\text{crop}})$ is based on a sum of pairwise interaction terms i.e.

$$\mathcal{L}_\partial(\mathbf{A}_{\text{crop}}) = \sum_{\{u,v\} \in \partial \mathbf{A}_{\text{crop}}} \mathcal{L}(u,v) \tag{16}$$

Lemma 3 allows us to first analyze $\mathcal{L}_\partial(\mathcal{B}_k)$ then use Lemma 3 for the final bound.

$$\mathcal{L}_\partial(\mathcal{B}_k) = \sum_{\{u,v\} \in \partial \mathcal{B}_k} \mathcal{L}(u,v) \tag{17}$$

$$\leq c|\partial \mathcal{B}_k| \max_{\{u,v\} \in \partial \mathcal{B}_k} \mathcal{L}(u,v) \tag{18}$$

using Lemma 2,

$$\leq c|\mathcal{B}_k|^{2/3} \max_{\{u,v\} \in \partial \mathcal{B}_k} \mathcal{L}(u,v) \tag{19}$$

Assuming [A1], [A2], and Lemma 1,

$$\leq c|T(\mathcal{B}_k)|^{2/3} \tag{20}$$

Combining this with Lemma 3 we have the final result

$$\frac{L_\partial(\mathbf{A}_{\text{crop}})}{T(\mathbf{A}_{\text{crop}})} = O((1 + r_{\text{cut}})T(\mathbf{A}_{\text{crop}})^{-1/3})$$

$\square$

As a reminder, this proposition implies that the boundary surface becomes less of an issue as the number of tokens grows.

(a)             (b)

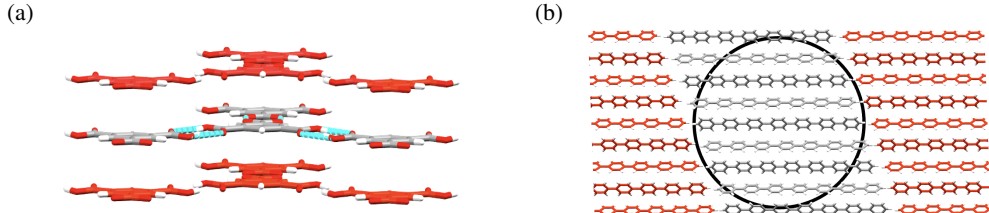

Figure 9: (a) $k$NN cropping as used in AlphaFold3 captures only the closest interactions (e.g. hydrogen bonds in trimesic acid, highlighted in blue) but does not capture more distant interactions that are equally crucial for crystallization (e.g. $\pi$-$\pi$ stacking in trimesic acid, highlighted in red). (b) centroid-based approaches will create anisotropic crops in elongated molecules, for example only capturing a 1-dimensional column (grey) in the ellipsoid $p$-quinquephenyl, while missing peripheral interactions (red).

## B    TECHNICAL DETAILS

Experiments were performed on heterogenous GPU clusters consisting of NVIDIA L40S and H100 GPUs. Models were primarily trained using multinode DDP training on L40S GPUs as cluster availability permitted. Inference is performed across all available hardware. Our model code is available to use here: https://github.com/OXtal/OXtal.

### B.1    TRAINING DATA PROCESSING

We curate training data from the Cambridge Structural Database (CSD, including releases up to May 1, 2025) under the following criteria: (i) 3D coordinates are present, (ii) the conventional $R$-factor $< 9\%$, (iii) the structure is derived from single crystal diffraction at ambient pressure, (iv) the structure is not polymeric, (v) the structure can be sanitized by RDKit with no missing heavy atoms, (vi) there is a known space group, and (vii) at most 250 non-hydrogen atoms are present per unit cell. To avoid leakage, we exclude any entry belonging to a test crystal family and any entry containing a molecular component that appears in test sets. To avoid oversampling certain clusters within each crystal family, near-duplicate polymorphs are collapsed using $\mathrm{RMSD}_{15} \leq 0.25$ Å, retaining the entry with the lowest $R$. Our final training dataset contains 594,202 crystals in total.

For data preparation, crystallographic disorder is resolved by selecting the disorder group with the highest occupancy. We remove hydrogens for training and extract kekulized bonds from crystallographic metadata. Prior to building supercells, molecules of the crystals are centered to lie within the unit cell, and the unit cells are transformed to their unique Niggli-reduced forms. Supercells are constructed by tile translation $\mathbf{T}_{ijk} = i\mathbf{a} + j\mathbf{b} + k\mathbf{c}$ for $i, j, k \in -1, 0, 1$ such that molecules with centroids inside the supercell boundary are included.

### B.2    ADDITIONAL MODEL DETAILS

#### B.2.1    STOICHIOMETRIC STOCHASTIC SHELL SAMPLING ($S^4$)

We provide the full algorithm for $S^4$ cropping in Algorithm 1. Recall that for a crystal $\mathcal{C}$, $\mathcal{M}$ denotes the set of molecules within the crystal, $\mathbf{A}$ denotes the tokenized atom array for all molecules $m \in \mathcal{M}$, and $X \subset \mathbb{R}^3$ denotes the Cartesian coordinates of each atom $a \in \mathbf{A}$. Note that for practical purposes, we assume $\mathcal{M}$ has a finite size. Additionally, $\mathcal{A}$ is the crystal's minimal *asymmetric unit*, and $d$ is a pre-computed intermolecular distance matrix, where $d(i, j) = \min_{x_i \in X(m_i), x_j \in X(m_j)} ||x_i - x_j||_2$.

Given a specified shell radius $r_{\text{cut}}$, we first sample a central molecule $m_c \sim \text{Uniform}(\mathcal{A})$. Then, we assign all other molecules $m_i \in \mathcal{M} \setminus m_c$ to a shell layer, as defined by $r_{\text{cut}}$. Note that each molecule $m_i$ can only belong to one shell layer. Next, with probability $(1 - p_{\max})$, we randomly sample how many shells to keep. We then add complete shells of molecules to our selected set until the maximum token budget $T_{\max}$ is reached. If no complete shell can fit within the token budget, we adaptively sample molecules in the first shell according to Algorithm 2 in order to preserve the stoichiometric ratios of the molecules in $\mathcal{A}$.

**Implementation details.** In practice, we begin by first considering a $3 \times 3 \times 3$ crystal supercell $\mathcal{C}^{(U)}$, where $U = 3I_3$. For each input crystal, we sample a molecule $m_c$ from the asymmetric unit of the central unit cell. After analysing the distribution of intermolecular distances for all crystals in the training set, we select $r_{\text{cut}} = 4.5$ Å (see Figure 10 and §D.2). Furthermore, in order to encourage the selection of larger $\mathbf{A}_{\text{crop}}$ blocks, we set $p_{\text{max}} = 0.8$ and $T_{\text{max}} = 640$.

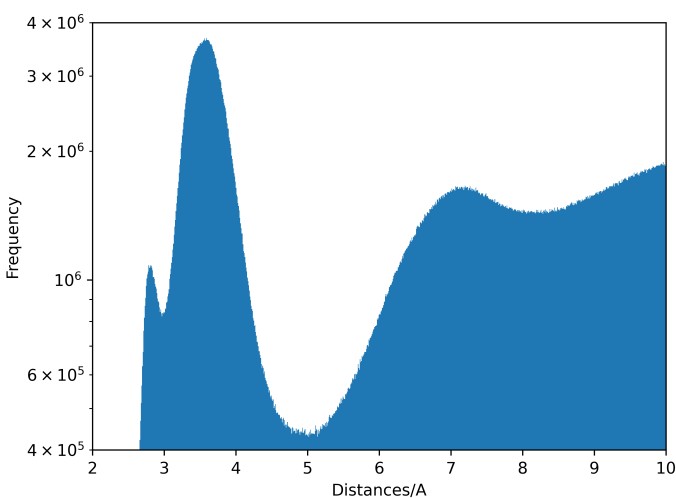

Figure 10: Truncated distribution of intermolecular distances in the processed training dataset.

---

**Algorithm 1** STOICHIOMETRIC STOCHASTIC SHELL SAMPLING

---

1: **Input:** $\mathcal{M}$, $\mathbf{A}$, $\mathcal{A}$, $d$, $r_{\text{cut}}$, $T_{\text{max}}$, $p_{\text{max}}$
2: $m_c \sim \text{Uniform}(\mathcal{A})$             ▷ Sample central molecule
3: $\mathcal{M} \leftarrow \mathcal{M} \setminus \{m_c\}$
4: Initialize $\mathcal{S} = \emptyset$, $k \leftarrow 1$
5: **while** $\mathcal{M} \neq \emptyset$ **do**             ▷ Compute shell layers around $m_c$
6:      $S_k \leftarrow \{m_i \in \mathcal{M} : d(m_c, m_i) \leq k \cdot r_{\text{cut}}\}$
7:      $\mathcal{S} \leftarrow \mathcal{S} \cup S_k$, $\mathcal{M} \leftarrow \mathcal{M} \setminus S_k$, $k \leftarrow k + 1$
8: **end while**
9:
10: $b_{\text{max}} \sim \text{Bernoulli}(p_{\text{max}})$             ▷ Sample maximum number of shells $S_{k_{\text{max}}}$ to keep
11: **if** $b_{\text{max}} = \text{False}$ **then**
12:      $k_{\text{max}} \sim \text{Uniform}\{1, \dots, k-1\}$
13:      $\mathcal{S} \leftarrow \{S_1, \dots, S_{k_{\text{max}}}\}$
14: **end if**
15:
16: Initialize $\mathbf{A}_{\text{crop}} \leftarrow \mathbf{A}(m_c)$, $i \leftarrow 1$,
17: **while** $|\mathbf{A}_{\text{crop}}| \leq T_{\text{max}}$ **do**          ▷ Add full integer shells within token budget $T_{\text{max}}$
18:      **if** $|\mathbf{A}_{\text{crop}}| + |\mathbf{A}(S_i)| > T_{\text{max}}$ **then**
19:          **break**
20:      **end if**
21:      $\mathbf{A}_{\text{crop}} \leftarrow \mathbf{A}_{\text{crop}} \cup \mathbf{A}(S_i)$, $i \leftarrow i + 1$
22: **end while**
23:
24: **if** $i = 1$ **then**          ▷ If no full shell fits, sample molecules according to stoichiometry
25:      $\mathbf{A}_{\text{crop}} \leftarrow$ ADAPTIVE STOICHIOMETRIC SAMPLING($\mathbf{A}$, $m_c$, $\mathcal{A}$, $T_{\text{max}}$, $S_1$)
26: **end if**
27: **return** $\mathbf{A}_{\text{crop}}$

---

---

**Algorithm 2** ADAPTIVE STOICHIOMETRIC SAMPLING

---

1: **Input: A**, $m_c$, $\mathcal{A}$, $T_{\max}$, $S$
2: Calculate proportion of each molecule type $p_t = \frac{|\{m_i \in \mathcal{A}:\text{type}(m_i)=t\}|}{|\mathcal{A}|}$
3: Initialize ideal targets: $R_t \leftarrow \text{round}(p_t \cdot |S|)$ for each type $t$
4: $\mathbf{A}_{\text{crop}} \leftarrow \mathbf{A}(m_c)$, $R_{\text{type}(m_c)} \leftarrow R_{\text{type}(m_c)} - 1$        $\triangleright m_c$ is already pre-selected
5:
6: **while** $|\mathbf{A}_{\text{crop}}| < T_{\max}$ **do**
7:      Bucket remaining molecules by type: $B_t = \{m \in S : \text{type}(m) = t\}$
8:      Compute adaptive weights for each type:
9:      **for all** $t \in T$ with $B_t \neq \emptyset$ **do**
10:          $w_t \leftarrow R_t/|B_t|$
11:      **end for**
12:      Normalize weights to probabilities: $P'_t = w_t / \sum_t w_t$
13:      Select type $t'$ randomly according to probabilities $P'_t$
14:      Sample $m' \sim \text{Uniform}(B_{t'})$
15:      **if** $|\mathbf{A}_{\text{crop}}| + |\mathbf{A}(m')| > T_{\max}$ **then**
16:          **break**        $\triangleright$ Stop if adding this molecule exceeds token budget
17:      **end if**
18:      $\mathbf{A}_{\text{crop}} \leftarrow \mathbf{A}_{\text{crop}} \cup \mathbf{A}(m')$
19:      $S \leftarrow S \setminus \{m'\}$, $B_{t'} \leftarrow B_{t'} \setminus \{m'\}$, $R_{t'} \leftarrow R_{t'} - 1$
20: **end while**
21: **return** $\mathbf{A}_{\text{crop}}$

---

### B.2.2 ARCHITECTURE

Our model adopts the AlphaFold3 (AF3) architecture via the public Protenix implementation (Team et al., 2025). We adopt Algorithm 5 from Abramson et al. (2024) to embed the reference conformer into the atom encoder. Atom-level tokens with relative position and entity encodings are processed by an AF3-style *Pairformer* trunk with recycling to produce single and pair representations; the multiple sequence alignment (MSA) pathways are disabled. These representations condition a structure-denoising head consisting of an atom-attention encoder, a transformer-based diffusion module, and an atom-attention decoder that outputs 3D coordinates.

### B.2.3 LOSSES

We use a combination of losses to encourage accurate structure prediction on both the global and local scales. Here we enumerate what the different losses are, how they are formulated, what they are useful for and how they lead to a correct diffusion model.

**SmoothLDDTLoss**. The Smooth local distance difference test (SmoothLDDT) loss is a smooth form of an existing LDDT loss (Mariani et al., 2013). The LDDT measures how well two structures align over all pairs of atoms at a distance closer than some predefined threshold. For us, this is $R_0 = 15\text{Å}$. These pairs of atoms form a set of local distances $L$. The LDDT score is the average of fractions at four distance thresholds: $[0.5, 1.0, 2.0, 4.0]$ Angstroms. The Smooth LDDT test, instead of using a binary test of whether or not the local distances are on the same side of the threshold, uses a sigmoid function. Let

$$L = \|x - x^T\| \tag{21}$$

$$L^{GT} = \|x^{GT} - (x^{GT})^T\| \tag{22}$$

$$\delta = \text{abs}(L - L^{GT}) \tag{23}$$

$$\epsilon = \frac{1}{4}\left[\text{sigmoid}(\frac{1}{2} - \delta) + \text{sigmoid}(1 - \delta) + \text{sigmoid}(2 - \delta) + \text{sigmoid}(4 - \delta)\right] \tag{24}$$

$$\text{mask} = \delta < 15 \tag{25}$$

Then the SMOOTHLDDT between two molecules of size $d$ is:

$$\text{SMOOTHLDDT}(x, x^{GT}) = \sum(\epsilon \cdot \text{mask})/(d(d-1)) \tag{26}$$

Our SmoothLDDT loss is then equal to

$$\mathcal{L}_{\text{sLDDT}}(x, x^{GT}) = \text{SMOOTHLDDT}(x, \text{ALIGN}(x^{GT}, x)) \tag{27}$$

where $\text{ALIGN}(a, b)$ performs an optimal rigid alignment of $a$ to $b$ in 3D.

### B.2.4 MODEL HYPERPARAMETERS

**Training**. Our training implementation is based off of the open-source Protenix model (Team et al., 2025). We disabled all MSA components. For training, we use an Adam optimizer with $\beta_1 = 0.9$, $\beta_2 = 0.95$, a learning rate of 0.0018, and a weight decay of 1e-8. The learning rate is additionally set to a scheduler with 1,000 warmup steps and a decay factor of 0.95 for every 50,000 steps. We report results for our model trained at 110,000 steps.

The atom encoder has 3 blocks with 4 heads. The atom embedding size is 128, whereas the pair embedding is size is 16. In terms of the main Pairformer trunk, we use 4 Pairformer blocks, each with 16 attention heads and a dropout rate of 0.25. The hidden dimension for the pair representation is 128, and the hidden dimension for the single representations are 384. Furthermore, we do 2 rounds of recycling. For the diffusion block, we use a diffusion batch size of 32 and a chunk size of 1. The actual diffusion transformer has 12 transformer blocks, with 8 attention heads. Crop size (and therefore token size for the diffusion transformer) is set to 640, with a 50/50 ratio of $S^4$ and $k$NN cropping for training. All components are set to seed 42.

**Inference**. At inference time, we use the following hyperparameters for our diffusion sampling: $\gamma_0 = 0.8$, noise scale lambda = 1.003, steps = 200, and step scale eta = 1.5. When generating samples, we generate 1 sample from 30 different seeds for each target crystal structure. Seeds are set from 0 to 29.

## C EXPERIMENTAL DETAILS

### C.1 SELECTION OF PERFORMANCE METRICS

The four metrics in the main text jointly probe physical plausibility ($\text{Col}_S$), rough packing similarity ($\text{Pac}_S$, $\text{Pac}_C$), intramolecular fidelity ($\text{Rec}_S$, $\text{Rec}_C$), and a holistic solve indicator ($\widetilde{\text{Sol}}_C$). Reporting both sample-level and crystal-level aggregates is essential: the former characterizes the distributional quality of all proposals, while the latter answers whether a target was recovered at least once. Concretely, crystal-level rates are an OR-aggregation over a target's samples. Evaluating both crystal and sample level metrics together helps diagnose failure modes such as mode collapse (high $\text{Pac}_C$ with low $\text{Pac}_S$) and low recall (the reverse).

**Thresholds and interpretability**. We adopt community conventions for $\text{RMSD}_k$ (Nessler et al., 2022):

- $\text{RMSD}_{15} < 1.0$ Å typically indicates the predicted packing reproduces the experimental match and lies in the exact energy basin as the experimental structure. This metric is the one used by the 5th CSP blind test (Bardwell et al., 2011).

- $1.0-2.0$ Å usually indicates the prediction shares the correct topology with mild lattice strain or small reorientations. If the H-bond graph, Z/Z', density, and PXRD are consistent, this is very likely recoverable to $< 1$ Å with a brief local relaxation. Otherwise, the structure is often structurally related to the ground truth (i.e. in or near the correct packing motif/topology or space group but with slip, molecular misorientation, or a cell mismatch). It demonstrates that the method can correctly identify the neighborhood of the global energy minimum, even if it can't pinpoint the exact minimum itself.

- $2.0-3.0$ Å typically indicate a significant mismatch. At this level of deviation, key intermolecular interactions, like hydrogen bonds, may be incorrect. The overall packing symmetry might also be different. However, the prediction might still capture a general feature of the packing (e.g., identifying a layered structure vs. a herringbone packing).

- above $3.0$ Å, these values are generally not considered useful. The deviation is so large that the predicted structure is almost certainly in a completely different and incorrect energy basin. Any similarity to the experimental structure is likely coincidental.

Our $\widetilde{\mathrm{Sol}}_C$ (collision-free, packing-similar, $\mathrm{RMSD}_{15} < 2\,\text{Å}$) is a thus a strict, early-stage indicator: not "solved," but reliably near-correct for downstream relaxation and re-ranking.

**Packing matching for non-periodic predictions**. Our generators output finite blocks without periodic conditions. For large inference blocks, it is possible to identify translation vectors that map the finite cluster onto itself (e.g., by correlating molecular centroids or local environments), least-squares fit three independent vectors to define a primitive lattice, and convert atoms to fractional coordinates. Nevertheless, CSD *COMPACK* can be directly used for robust alignment of a block against a crystal structure while preserving standard CSP rigor. First, we employ standard practices of avoiding hydrogen-atom alignment and allowing mismatches in connectivity annotations (e.g., bond orders). In the absence of periodic images, greedy pruning can miss valid correspondences; therefore for all methods (including DFT baselines) we use a search time of 10 seconds with a distance and angle threshold of 50%. This enables reliable COMPACK outputs on non-periodic inputs *without diluting the criterion*: acceptance still hinges on multi-molecule superposition (15-molecule cluster, $\geq 8$ matched) and the same $\mathrm{RMSD}_k$ thresholds used in periodic CSP benchmarks. Lastly, we note that it is possible to deduce the underlying Bravais lattice of a large periodic point cloud by analyzing the set of interatomic difference vectors (a Patterson analysis): in the resulting Patterson function, lattice translation vectors appear as a high-multiplicity, regularly spaced subset of the interatomic vectors, from which the lattice parameters (and hence a primitive cell) can be recovered.

### C.2 Molecular CSP on Rigid and Flexible Datasets

We construct our rigid and flexible test datasets using crystals from the Cambridge Crystal Structure Database (CCDC), following the same processing procedure outlined in §B.1. Both rigid and flexible datasets are comprised of 50 molecular crystals. These are generally crystals with practical relevance or newly-released crystals. We generally define rigid molecules to contain 0-3 rotatable bonds, and flexible molecules to contain more than that. We note there is a small nuance in flexibility, as we refine it in the molecular context (by ring restriction or number of known polymorphs, etc.), this means crystals such as rubrene (QQQCIG), tetraphenylporphryin (TPHPOR), CL-20 (PUBMUU) are defined as "rigid;" ROY (QAXMEH), galunisertib (DORDUM), sulfathizaole (SUTHAZ), and flufenamic acid (FPAMCA) are defined as "flexible." The exact CSD identifiers are provided below for reference.

Molecules in the rigid dataset: XULDUD, GUFJOG, QAMTAZ, BOQQUT, BOQWIN, UJIRIO, PAHYON, XATMIP, HAMTIZ, XATMOV, AXOSOW, SOXLEX, WIDBAO, HXACAN, ACSALA, PUBMUU, GLYCIN, TEPNIT, QQQCIG, ZZZMUC, FOYNEO, ANTCEN, URACIL, BTCOAC, CORONE, GUJTOX, TPHPOR, ACETAC, IMAZOL, CEBYUD, CILJIQ, CUMJOJ, DEZDUH, DOHFEM, GACGAU, GOLHIB, HURYUQ, IHEPUG, LECZOL, NICOAM, ROHBUL, UMIMIO, WEXREY, CAPRYL, ACACPD, BPYRUF, JIVNOV, MUBXAM, VUJBUB, NAVZAO.

Molecules in the flexible dataset: QAXMEH, DORDUM, BOTHUR, MOVZUW, SUTHAZ, TPPRHC, FPAMCA, INDMET, MELFIT, YIGPIO, TEHZIP, DMANTL, YIGDUP, UWEQUL, COWZIA, FUVLAN, RUTJUP, MUYRIL, QUTZUE, MURQUP, CUWLIT, QURPUS, DUZTOL, VUZQUG, MUZKOL, YUYLAJ, BUYWUR, HABMUX, QURBEO, HUTDOT, CUYGEM, MUSTIH, MUSXUX, DABHIC, FACDOH, RUWFAU, CAFHUR, HUWPUO, CAFHEB, HUTKOA, DUVCOQ, QUXHAW, CABFIZ, QUSQAA, MUSYAE, CUYVUR, JABXIY, RUSXEM, CUYWAY, YABFAN.

### C.3 *Ab initio* Machine Learning Baselines

We compare OXTAL against three *ab initio* machine learning baselines: (i) a lightweight transformer trained with flow matching on our dataset (§C.3.1), (ii) the publicly released ASSEMBLEFLOW-ATOM model (Guo et al., 2025), and (iii) ALPHAFOLD3 (Abramson et al., 2024) used as a generative baseline. Our complete table of results are reported in Table 3 on both rigid and flexible molecular CSP benchmarks using $n = 30$ samples per crystal target.

### C.3.1 A-Transformer (ours)

**Model.** A-Transformer is a lightweight transformer encoder operating on atom tokens with unit-cell features. The model is a PyTorch `nn.TransformerEncoder` with hidden size $d_h = 512$,

Table 3: Performance of *ab-initio* machine learning models, including AlphaFold3, on the Rigid molecular CSP dataset. For each model, results are calculated using $n = 30$ samples for each crystal target.

| Model | $\text{Col}_S \downarrow$ | $\text{Pac}_S \uparrow$ | $\text{Pac}_C \uparrow$ | $\text{Rec}_S \uparrow$ | $\text{Rec}_C \uparrow$ | $\widetilde{\text{Sol}}_C \uparrow$ |
|---|---|---|---|---|---|---|
| A-Transformer | 0.731 | 0.015 | 0.060 | 0.033 | 0.120 | 0.060 |
| AssembleFlow | 0.524 | 0.001 | 0.040 | 0.211 | 0.760 | 0 |
| AlphaFold3 | 0.114 | 0.089 | 0.340 | 0.133 | 0.480 | 0 |
| OXTAL | 0.011 | 0.873 | 1.000 | 0.737 | 0.960 | 0.300 |

$L = 13$ layers, and $H = 4$ heads. Each atom embedding includes element and charge information, time $t \in (0, 1]$, molecular fingerprints, and, when enabled, unit-cell lengths $(a, b, c)$ and angles $(\alpha, \beta, \gamma)$. Two output heads are used: a coordinate head ($\mathbb{R}^3$ per token) and, optionally, a rotation head (unit quaternion, disabled in our main runs).

**Training.** The objective is rigid-cluster translation flow matching in $\mathbb{R}^3$. We linearly interpolate between ground-truth translations $s_0$ and random in-cell starts $s_1$,

$$s_t = (1 - t)\, s_0 + t\, s_1, \quad t \sim \text{Unif}[t_{\min}, 1],$$

and predict denoised coordinates $\hat{x}_0$. The loss is an $x_0$ regression term:

$$\mathcal{L} = \lambda_{\text{trans}} \sum_i \|x_0^{(i)} - \hat{x}_0^{(i)}\|^2.$$

**Inference and Evaluation.** At inference we integrate from $t=1$ to $t_{\min}$ with Euler steps, producing clusters in a P1 box. This baseline is deliberately minimal: it ignores rotation flow matching and lattice prediction, relying solely on translation flow matching and unit-cell features. It provides a capacity-matched reference against which to measure the benefits of OXTAL's architecture and training.

### C.3.2 ASSEMBLEFLOW

We evaluate the newly released ASSEMBLEFLOW-ATOM model (Guo et al., 2025) using the authors' checkpoints without re-training or fine-tuning. For each molecule we generate clusters of 15 rigid copies from an RDKit ETKDG conformer, applying random rotations and translations with uniform offsets $[-S, S]^3$, where $S \in \{10, 15, 20\}$ Å. Three seeds $\{42, 7, 2024\}$ and seven checkpoints yield 63 runs per molecule. To standardize evaluation, we randomly sample 30 outputs per crystal and compute metrics on those. AssembleFlow enforces rigid-molecule assembly and provides a strong baseline for rigid-packing quality, but is unable to handle flexible molecules, as reflected in the results presented in Table 3.

### C.3.3 ALPHAFOLD3

We additionally evaluate ALPHAFOLD3 as a structural generative baseline for the Rigid molecule dataset. For each target molecule, we generate conformations using default protein–ligand generative settings, treating each conformer as a candidate crystal packing. Inference is performed based on 30 counts of the molecule, and thirty samples per target are evaluated in the same way as for other baselines. While ALPHAFOLD3 was not designed for crystal structure prediction, including it highlights the gap between general-purpose biomolecular structure generators and CSP-specific models.

### C.4 CCDC CSP BLIND TEST DETAILS

To contextualize our work, we provide a summary of the 5th, 6th, and 7th CCDC CSP blind tests. These community-wide challenges have benchmarked the state-of-the-art in predicting the crystal structures of organic molecules from their chemical graphs alone, primarily relying on density functional theory (DFT). Here, we provide brief descriptions of these tests, and refer the audience to

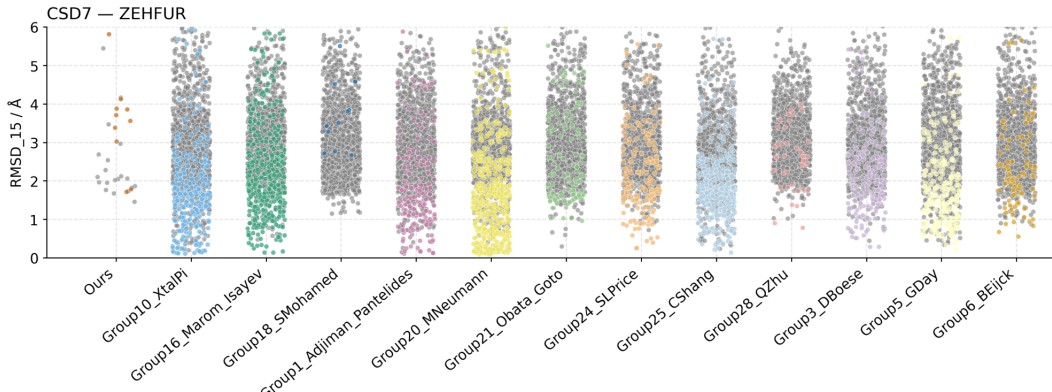

Figure 11: Beeswarm plot of submitted structures for Molecule XXXI (ZEHFUR) from CSD7. In 30 inference samples, OXTAL obtains performance comparable to some DFT groups with thousands of submitted samples. Colored points signify a partial packing similarity and gray points signify packing mismatch (e.g. 3 out of 15 molecules match with low RMSD, i.e. overall packing is significantly different).

Bardwell et al. (2011); Reilly et al. (2016); Hunnisett et al. (2024) for details. Performance is assessed using the COMPACK algorithm, which quantifies structural similarity through root mean square deviation (RMSD) of molecular clusters.

In the first four CSP blind tests, the field evolved from force-field landscapes to the first widespread use of periodic dispersion-corrected DFT (DFT-D), which proved decisive in 4th blind test and set the stage for DFT to become the de facto standard for final lattice-energy evaluation. The 5th blind test (Bardwell et al., 2011) marked a significant increase in the complexity of the target molecules. The six targets included rigid molecules, semi-flexible molecules, a 1:1 salt, a highly flexible pharmaceutical-like compound, and two co-crystals. This test highlighted what has become a standard workflow in CSP: broad structure generation (e.g., grid or quasi-random sampling, parallel tempering), followed by local minimization and hierarchical re-ranking. While at least one successful prediction was submitted for each target, the success rate was lower than in previous tests, underscoring the increased difficulty. DFT-D demonstrated its reliability for discriminating between competing structures for small and moderately flexible molecules, but handling high flexibility and complex solid forms remained a major challenge. A total of 15 groups submitted structures for this blind test, although the report contains only 14 groups.

The 6th blind test (Reilly et al., 2016) continued with five challenging targets: a small, nearly rigid molecule; a polymorphic former drug candidate; a chloride salt hydrate; a co-crystal; and a large, flexible molecule. This test solidifed the "search–optimize–rank" pipeline. On the search side, more than half of the methods allowed intramolecular flexibility during exploration, and many adopted hierarchical filtering starting with generating conformer and packing, followed by progressively tighter optimization and pruning. In optimization and ranking, dispersion-corrected periodic DFT became mainstream, with van der Waals (vdW) models (e.g., D3/D3(BJ), MBD) and multipole-based electrostatics or SAPT-derived potentials used to refine close competitors. All experimental structures were predicted by at least one submission except a potentially disordered $Z' = 2$ polymorph. The results demonstrated that while DFT-D provided reliable baseline energetics, accurate treatment of conformational flexibility remained the primary bottleneck. A total of 25 groups participated in this blind test.

The 7th blind test (Hunnisett et al., 2024) introduced a two-phase structure to separate structure generation from ranking challenges. The seven targets featured a silicon-iodine containing molecule, a copper coordination complex, a near-rigid molecule, a co-crystal, a polymorphic small agrochemical, a highly flexible polymorphic drug candidate, and a polymorphic morpholine salt. The test also featured one of the most challenging systems in the history of the blind tests: a large, highly polymorphic pharmaceutical drug candidate. A key finding from the structure generation phase was that while different search methods often identified overlapping sets of low-energy structures,

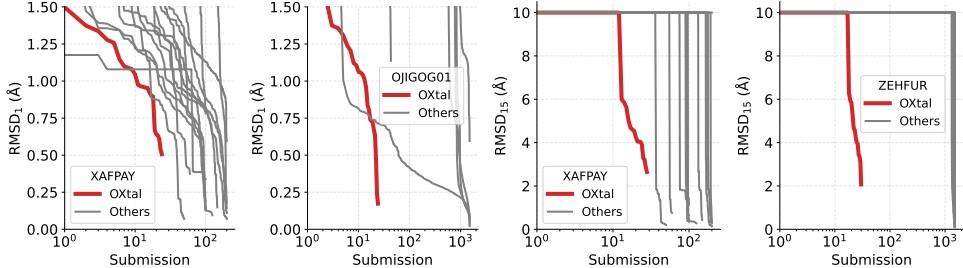

Figure 12: Sample efficiency plots for XAFPAY (blind test 6, a flexible molecule), OJIGOG01 (blind test 7), and ZEHFUR (blind test 7). OXTAL is able to quickly recover both molecular conformers ($RMSD_1$) and periodicity ($RMSD_{15}$).

their success rates tended to decrease as the complexity of the target molecule increased. For ranking, periodic dispersion-corrected GGA DFT methods produced results in excellent agreement with experiments across most targets, whereas higher-level corrections, full free-energy treatments, and ML on DFT potentials were less decisive for these specific systems than expected. Additionally, dynamic and static disorder (and high Z') remained difficult. A total of 22 groups participated in the structure generation phase of this blind test.

These blind tests shows a clear trend that DFT-based ranking has become reliable, but limitations persist: computational expense, difficulty capturing kinetic effects, challenges with disorder and flexibility and high Z' number, and the fundamental limitation that thermodynamic ranking often fails to predict which polymorphs are experimentally observable. Our approach aims to address these limitations by directly learning both thermodynamic and kinetic regularities from data, with the hope of eventually eliminating the need for extensive search, local optimization, and post-hoc ranking altogether. By generating a small number of high-quality structures directly, we bypass the traditional generate-optimize-rank pipeline that has dominated CSP. Tables 8, 9, and 10 in §I show comparisons for all submitted structures by each group (sans ranking)The beeswarm plot (Figure 11) of submissions for molecule XXXI further highlights the extensive sampling currently required by DFT methods compared to OXTAL. This point is further emphasized in Figure 12, which plots the best $RMSD_{1/15}$ achieved at a given $n$ submission size for a few blind test crystals.

**Result computation**. Because each submission group can choose which crystals to attempt, in §4.2 we report metrics that are averaged over submission groups. For sample-level metrics, we first compute each rate separately for every group, then take the mean of these per-group rates across all groups. $n_S$ is the average number of samples per group per target. For crystal-level metrics, we average the rates over each group, and across *all* available targets for that competition. We count groups who did not submit any submissions for a crystal as a miss, since they did not show evidence of successfully solving it.

Note that in blind test 7 (Hunnisett et al., 2024), Target XXX has two forms with different stoichiometries (MIVZEA and MIVZIE). For evaluation purposes, *ab initio* machine learning methods attempted 30 structures for each stoichiometry. As such, per-sample metrics for competition 7 are computed with these additional samples, and we consider 8 total targets for per-crystal metrics.

### C.4.1    INFERENCE TIME COST

We compute computational costs by multiplying the reported wall–clock compute time by an AWS on–demand unit price with hours taken directly from the three blind tests references (Bardwell et al., 2011; Reilly et al., 2016; Hunnisett et al., 2024). Because the papers report heterogeneous processors and often normalize times to ∼3.0 GHz core–hours, we treat one normalized CPU hour as one AWS vCPU–hour; we then price CPU time at the c5.large on–demand rate in Sept. 2025, i.e., \$0.085 per *2* vCPUs ⇒ \$0.0425 per vCPU–hour, so CPU cost = hours × \$0.0425. For L40S GPU runs, we map directly to g6e.xlarge instances (which contains one NVIDIA L40S) and are priced at \$1.861 per instance–hour, so GPU cost = hours × \$1.861. We apply the rates pro–rata without further rounding. All prices are on–demand EC2 instances as of September 2025. For reference, exact reported times used are presented in Tables 11, 12, and 13.

# D  MODEL ABLATION STUDIES

Here, we provide a series of different model ablation studies. Due to the high computational demand of training OXTAL, all results in this section are reported at 50k training steps instead of the 110k reported in the main body of the paper. Following our predefined evaluation protocol, we evaluate $n_S = 30$ generated samples with 30 molecular copies for each crystal target.

## D.1  CROPPING METHOD ABLATION

In order to isolate the performance gain from $S^4$, we directly compare it against other standard cropping methods. Overall, $S^4$ performs the best across all metrics and datasets with centroid radius performing the second best, indicating that there may not be many oblong molecules in the evaluation datasets. However, these results support the benefits of using $S^4$, since it is not subject to the same pitfalls of $k$NN and centroid radius cropping as illustrated in Figure 9.

Also, note that our implementation of centroid radius cropping is more generous than the standard one used in the literature (Abramson et al., 2024). Specifically, naive radius cropping selects any atom that lies within a pre-defined spatial radius. However, this often results in partial molecules being cropped, which naturally leads to lower performance. Instead, we consider all atoms in a given molecule if its *centroid* lies within 15 Å of a randomly sampled central molecule, and only consider complete molecules up to our token budget, which mitigates the issue of fragmented molecules during training. $k$NN is also implemented such that it only considers complete molecules up to the token budget, ordered by neighbor distance from a randomly sampled central molecule.

Table 4: Results for different cropping methods with best results **bolded** and second best underlined.

| Crop Method | Dataset | $\mathrm{Col}_S \downarrow$ | $\mathrm{Pac}_S \uparrow$ | $\mathrm{Pac}_C \uparrow$ | $\mathrm{Rec}_S \uparrow$ | $\mathrm{Rec}_C \uparrow$ | $\widetilde{\mathrm{Sol}}_C \uparrow$ |
|---|---|---|---|---|---|---|---|
| $k$NN | Rigid | 0.086 | 0.631 | 0.920 | 0.520 | 0.780 | 0.220 |
| | Flexible | 0.444 | 0.123 | 0.700 | 0 | 0 | 0.020 |
| | CSP5 | 0.085 | 0.470 | 0.667 | 0.433 | 0.833 | 0 |
| | CSP6 | 0.132 | 0.353 | 0.800 | 0.007 | 0.200 | **0.200** |
| | CSP7 | 0.312 | 0.165 | **0.750** | 0.005 | 0.125 | 0 |
| Centroid Radius | Rigid | 0.040 | 0.669 | **0.980** | 0.591 | **0.940** | **0.340** |
| | Flexible | 0.223 | 0.184 | 0.720 | **0.033** | 0.300 | 0.040 |
| | CSP5 | 0.054 | 0.470 | **0.833** | 0.429 | 0.833 | 0 |
| | CSP6 | 0.151 | 0.324 | **1.000** | **0.173** | 0.400 | 0 |
| | CSP7 | 0.104 | **0.225** | **0.750** | **0.104** | 0.250 | 0 |
| Ours ($S^4$) | Rigid | **0.026** | **0.688** | 0.940 | **0.629** | **0.940** | 0.280 |
| | Flexible | **0.143** | **0.260** | **0.800** | 0.031 | **0.360** | **0.140** |
| | CSP5 | **0.000** | **0.500** | **0.833** | **0.478** | **1.000** | 0 |
| | CSP6 | **0.047** | **0.440** | **1.000** | 0.107 | **0.800** | **0.200** |
| | CSP7 | **0.033** | 0.221 | 0.625 | **0.104** | **0.500** | 0 |

## D.2  $S^4$ RADIUS SIZE ABLATION

Next, we investigate the sensitivity of OXTAL to the choice of $S^4$ shell radius size. From our results in Table 5, we see that there are generally minimal changes in performance across three different choices of radius size. Recall that our main OXTAL model uses a shell radius size of 4.5, which was selected from analyzing the distribution of intermolecular distances presented in Figure 10. From this figure, both 4.5 and 5 Å are natural choices for the first shell, however due to the "layered" nature of crystallization, we consider a second shell at 9 rather than 10 Å to be more fitting.

Overall, our current implementation of $S^4$ may slightly favor smaller shell radius sizes due to the overall token budget constraint, which limits the number of full shells we are able to accommodate. Regardless, we see that all three radius settings for $S^4$ outperform existing cropping methods from Table 4, suggesting that OXTAL is somewhat robust to reasonable choices in $S^4$ radius size.

Table 5: Effect of $S^4$ shell radius size, with best results **bolded** and second best underlined.

| Radius Size | Dataset | $\text{Col}_S \downarrow$ | $\text{Pac}_S \uparrow$ | $\text{Pac}_C \uparrow$ | $\text{Rec}_S \uparrow$ | $\text{Rec}_C \uparrow$ | $\widetilde{\text{Sol}}_C \uparrow$ |
|---|---|---|---|---|---|---|---|
| | Rigid | **0.015** | **0.694** | **0.980** | 0.596 | 0.920 | **0.320** |
| | Flexible | **0.109** | **0.290** | **0.820** | **0.049** | **0.380** | **0.140** |
| 3.5 Å | CSP5 | 0.006 | 0.439 | 0.833 | 0.433 | 0.833 | 0 |
| | CSP6 | 0.068 | 0.409 | **1.000** | **0.174** | 0.600 | **0.400** |
| | CSP7 | 0.095 | **0.238** | **0.750** | 0.110 | **0.500** | 0 |
| | Rigid | 0.026 | 0.688 | 0.940 | **0.629** | 0.940 | 0.280 |
| | Flexible | 0.143 | 0.260 | 0.800 | 0.031 | 0.360 | **0.140** |
| 4.5 Å | CSP5 | **0.000** | 0.500 | 0.833 | 0.478 | **1.000** | 0 |
| | CSP6 | **0.047** | 0.440 | **1.000** | 0.107 | **0.800** | 0.200 |
| | CSP7 | **0.033** | 0.221 | 0.625 | 0.104 | **0.500** | 0 |
| | Rigid | 0.047 | 0.690 | 0.940 | 0.625 | **0.960** | 0.220 |
| | Flexible | 0.216 | 0.241 | 0.740 | 0.021 | 0.300 | 0.080 |
| 5.5 Å | CSP5 | 0.012 | 0.470 | 0.667 | 0.439 | 0.667 | 0 |
| | CSP6 | 0.147 | 0.375 | **1.000** | 0.147 | 0.600 | 0.200 |
| | CSP7 | 0.101 | 0.234 | **0.750** | **0.142** | 0.375 | 0 |

## D.3 Model Size Ablation

To further investigate the effect of model size, we train a smaller version of OXTAL that contains roughly 50M total parameters, which is half the size of the one we report in the main paper.

Table 6: Effect of OXTAL model size, with best results highlighted in **bold**.

| Model | Dataset | $\text{Col}_S \downarrow$ | $\text{Pac}_S \uparrow$ | $\text{Pac}_C \uparrow$ | $\text{Rec}_S \uparrow$ | $\text{Rec}_C \uparrow$ | $\widetilde{\text{Sol}}_C \uparrow$ |
|---|---|---|---|---|---|---|---|
| | Rigid | 0.731 | 0.015 | 0.060 | 0.033 | 0.120 | 0.060 |
| | Flexible | 0.900 | 0.001 | 0.020 | 0 | 0 | 0.020 |
| A-Transformer | CSP5 | 0.833 | 0 | 0 | 0 | 0 | 0 |
| | CSP6 | 0.967 | 0 | 0 | 0 | 0 | 0 |
| | CSP7 | 0.950 | 0 | 0 | 0 | 0 | 0 |
| | Rigid | 0.524 | 0.001 | 0.040 | 0.211 | 0.760 | 0 |
| | Flexible | 0.883 | 0 | 0 | **0.021** | 0.140 | 0 |
| AssembleFlow | CSP5 | 0.717 | 0 | 0 | 0.150 | 0.500 | 0 |
| | CSP6 | 0.800 | 0 | 0 | 0.073 | 0.200 | 0 |
| | CSP7 | 0.808 | 0 | 0 | 0.063 | **0.250** | 0 |
| | Rigid | **0.066** | **0.712** | **0.980** | **0.621** | **0.940** | **0.320** |
| | Flexible | **0.281** | **0.221** | **0.700** | 0.017 | **0.240** | **0.060** |
| OXTAL (50M) | CSP5 | **0.170** | **0.491** | **1.000** | **0.400** | **0.833** | 0 |
| | CSP6 | **0.261** | **0.370** | **1.000** | **0.094** | **0.400** | **0.200** |
| | CSP7 | **0.159** | **0.173** | **0.750** | **0.105** | **0.250** | **0.125** |

As shown in Table 6, the smaller 50M parameter version of OXTAL still outperforms existing *ab-initio* ML methods, which we have provided again here for reference. This result reinforces the benefits of OXTAL's lattice-free and non-equivariant architecture at a smaller scale.

## E Conformer Analysis

We now investigate the influence of the feature molecular conformer used by OXTAL. For each molecule, the model is conditioned on (i) its bond information and (ii) a 3D structure obtained using RDKit ETKDG followed by relaxation with the semi-empirical quantum-chemical method GFN2-xTB. This conformer serves solely as a feature conditioning signal for the computational

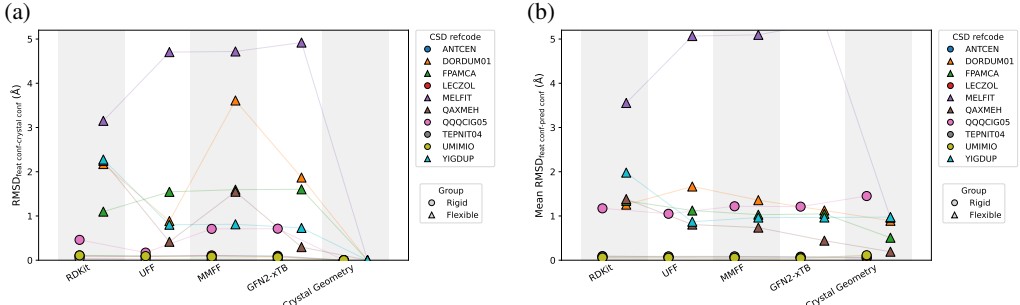

Figure 13: Analysis of input and final conformer in ten crystals for which OXTAL obtains $RMSD_1 <$ 0.5Å. (a) Different input conditioning conformers and their differences with the ground truth. Generally, flexible molecules' conformers have significant deviations to the crystal geometry. (b) Different input conditioning conformers and their differences with the final, model-predicted conformer. Generally, the model final conformers that are significantly from the conditioning.

prior, rather than as an experimental oracle: the generative process always starts denoising from random atomic coordinates, with the conformer providing only a reference for physically plausible initialization derived from quantum-chemical approximations as a feature input. To quantify how this initialization influences generation quality, we perform the following targeted analyses:

1. Distance of the feature conformer relative to ground truth. In Figure 13(a), we show that the feature conditioning conformers for flexible molecules can be significantly different than the ground truth crystal conformation. This shows how often the model must depart from or refine the initial geometry.

2. Distance of the feature conformer relative to the prediction. In Figure 13(b), we show that there are significant differences between the feature conformer with the final, model-predicted conformer. Structures predicted by the diffusion process is different than the one conditioned in the input embedding, i.e. it is not merely reproducing the structure used in input embedding.

3. Robustness to feature conformer quality. We assess robustness by replacing the GFN2-xTB initialization with alternative or deliberately perturbed conformers,

   - RDKit-ETKDG conformers
   - Universal forcefield conformers
   - MMFF94 conformers
   - GFN2-xTB conformers
   - Ground truth crystal conformers.

   In Figure 14(b), we plot generation performance (e.g., $RMSD_{15}$) as a function of input distortion level. We also plot the best $RMSD_{15}$ for each case in Figure 14(a). These plots show that the model's predicted crystal packing is not significantly affected by the different conformers or their differences with the ground truth crystal conformation.

4. The issue of $Z'$. When $Z' > 1$, the asymmetric unit contains several symmetry-independent molecules. These molecules can have similar conformation (but in different orientation) or genuinely different conformers. $\sim 10\%$ of the training set contains such examples, and in this subset, mean $Z'$ is 2.14. OXTAL conditions every molecular copy with the same feature conformer, and can occasionally approximately solve crystals with $Z' > 1$ (e.g. Target XXIII in CSP 6, XAFPAY02, $RMSD_{15} = 1.91$Å.)

Together, these analyses provide a clear and quantitative view of how the initial conformer affects OXTAL: how different it is from the target, how effectively the model corrects it, how robust generation is to poor or perturbed inputs, and whether supplying multiple conformers per $S^4$ unit impacts performance.

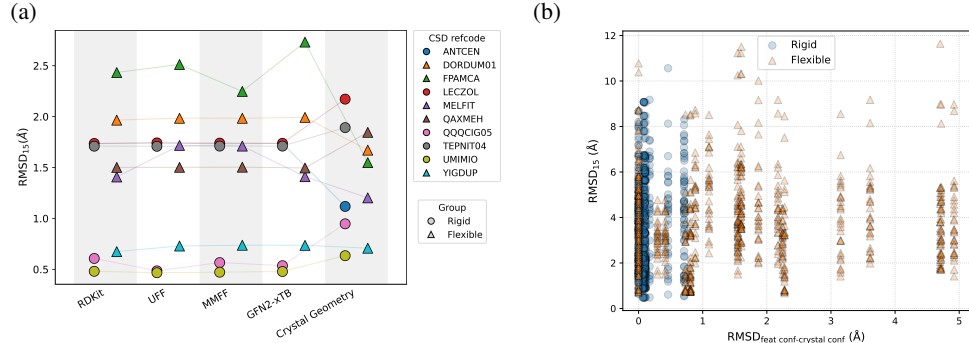

Figure 14: Analysis of input conformer with the final predicted crystal packing. (a) Best $RMSD_{15}$ from 30 samples per target when using different initial conformers. Generally, the model is robust against different sources of the input conditioning conformer. (b) All $RMSD_{15}$ across ten targets relative to the difference in input conditioning and crystal conformer differences. This shows the model can still produce good packings even if the input conditioning conformer there is very different from the crystal conformation

## F INFERENCE ANALYSIS

In this section, we provide a deeper analysis into the inference and generation capabilities of OXTAL. Specifically, we investigate the performance of OXTAL on generating significantly larger crystal blocks, evaluate how the model performs when allowing for additional sampling, and provide a small qualitative analysis on the diversity of generated samples.

### F.1 INFERENCE ON LARGER CRYSTAL BLOCKS

To evaluate the long-range periodicity of OXTAL, we perform inference on increasingly larger crystal blocks for a handful of different crystals (Figure 15). The respective number of atoms, and therefore tokens, for each molecule are provided in parentheses as follows: CAPRYL (10), CUYVUR (18), HURYUQ (14), QQQCIG05 (42), UMIMIO (12). Recall that OXTAL is trained with a maximum token budget of 640 (§B.2.4), hence generating 200 copies of CAPRYL requires 2,000 tokens and 100 copies of QQQCIG05 requires 4,200 tokens, which are far beyond anything the training dataset would have seen.

In general, conformer recovery (denoted by $RMSD_1$) remains somewhat constant as more molecule copies are generated. This makes sense because if the model is already able to capture the correct crystal conformer, generating more copies of that should not change the conformer itself significantly. In terms of lattice periodicity (denoted by $RMSD_{15}$), OXTAL does struggle slightly with generating more molecules, since the packing arrangements cover distances that are further away. However, we note that although the $RMSD_{15}$ does increase for these larger blocks, they still mostly remain under the 2.0 Å threshold to be considered "approximately solved."

This analysis supports the long-range generalizability of OXTAL to generate larger periodic packings, and we provide a visualization of the approximately solved larger packing for ANTCEN in Figure 16, which contains over 2,400 tokens.

### F.2 ADDITIONAL SAMPLE EVALUATION FOR CSP BLIND TEST 5

For all previously reported results, we evaluate OXTAL using 30 generated samples per crystal target. This is done in order to highlight the sample efficiency of OXTAL (cf. Figures 5 and 6), since few-sample success is critical for downstream screening and design. However, we note that this puts OXTAL at somewhat of a disadvantage when comparing against traditional DFT methods, which often generate hundreds or thousands of candidate packings. Here, we provide a more direct comparison of OXTAL to DFT methods by evaluating OXTAL using the same number of generated samples on the CSP Blind Test 5 dataset.

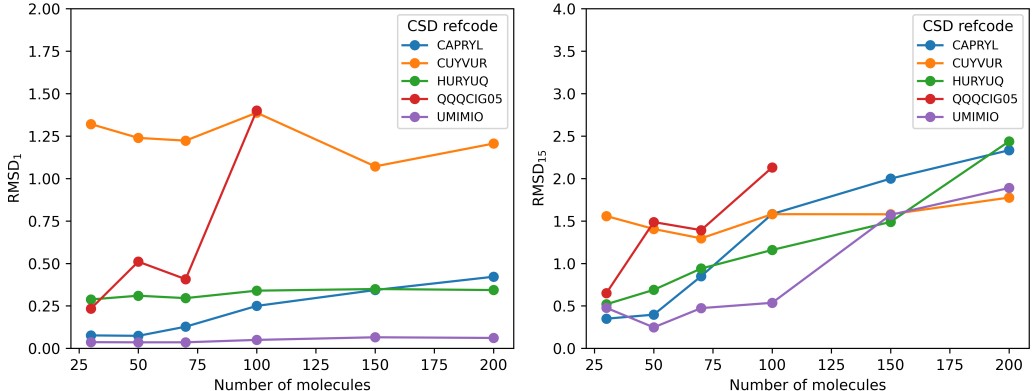

Figure 15: Performance of OXTAL on increasingly larger inference crystal blocks (denoted by the number of molecule copies generated). $\text{RMSD}_1$ (left) remains roughly constant, whereas $\text{RMSD}_{15}$ increases slightly for larger predicted packings. Note that QQQCIG05, which contains 42 heavy atoms (i.e. tokens) per molecules, goes out of memory past 100 molecules.

Table 7: Evaluation of OXTAL on CSP Blind Test 5 with the same number of generated samples per crystal target ($n_S$) as $\text{DFT}_{\text{avg}}$.

| Model | $n_S$ | $\text{Col}_S \downarrow$ | $\text{Pac}_S \uparrow$ | $\text{Pac}_C \uparrow$ | $\text{Rec}_S \uparrow$ | $\text{Rec}_C \uparrow$ | $\widetilde{\text{Sol}}_C \uparrow$ |
|---|---|---|---|---|---|---|---|
| $\text{DFT}_{\text{avg}}$ | 464 | 0.039 | 0.323 | 0.611 | **0.784** | 0.733 | **0.544** |
| OXTAL | 30 | **0.006** | **0.667** | 0.833 | 0.572 | 0.833 | 0.167 |
| OXTAL | 500 | 0.009 | 0.536 | **1.000** | 0.548 | **1.000** | 0.500 |

As expected, allowing OXTAL to generate more candidate packings increases its performance on our per-crystal evaluation metrics, while remaining roughly the same on per-sample metrics. Notably, we also see that increasing the number of generated samples bring the approximate solve rate of OXTAL equal to that of $\text{DFT}_{\text{avg}}$. This suggests that the model sampler is indeed able to generate accurate crystal packings and compete with traditional DFT methods.

## G  ADDITIONAL RELATED WORK

**Physical approaches to crystal structure prediction**. Physical approaches mostly rely on search and sampling using an energy function. Some classical methods infused domain knowledge, such as starting with initial guesses like a unit cell, and employed varying parameters, random sampling (Case et al., 2016; Pickard & Needs, 2011; Tom et al., 2020), guidance from force-fields (van Eijck, 2002), or constructed guesses based on chemical principles (Ganguly & Desiraju, 2010). Recent methods have also applied more structured search algorithms, such as simulated annealing (Reinaudi et al., 2000; Earl & Deem, 2005), genetic algorithms (Curtis et al., 2018; Lyakhov et al., 2013), particle swarm optimization (Wang et al., 2010), and basin-hopping (Banerjee et al., 2021).

**ML potentials**. Computational complexity of DFT and availability of large datasets (Levine et al., 2025; Sriram et al., 2024; Barroso-Luque et al., 2024; Smith et al., 2020) enabled the development of universal machine learning interatomic potentials (MLIP) (Gasteiger et al., 2021; 2022; Batatia et al., 2022; Liao et al., 2024; Batatia et al., 2025; Wood et al., 2025) to predict energy and forces of different atomistic systems (from small molecules to inorganic crystals) at a fraction of DFT costs. The next generation of physical approaches to CSP replaced DFT with MLIPs and showed benefits in material discovery (Merchant et al., 2023) with the recent FastCSP model (Gharakhanyan et al., 2025) applied to organic CSP.

**Generative models for inorganic crystal structure prediction**. Generative models have emerged as a promising new paradigm for crystal structure prediction, focusing mainly on conformer search in molecular structures as well as inorganic, periodic crystals. For inorganic crystal structures,

which consist of a periodic unit cell of atoms, generative models were first applied for unconditional de-novo generation of new crystals (Xie et al., 2022) and later used for structure generation conditioned on crystal composition (Jiao et al., 2023; 2024; Miller et al., 2024; Levy et al., 2025). The use of generative models has spanned multiple modeling methods, including diffusion models with equivariant models (Jiao et al., 2023; 2024), symmetry-aware diffusion models (Levy et al., 2025), and flow-matching models on specialized manifolds (Miller et al., 2024). Recent work have also applied large-language models to crystal generation and structure prediction with more varied success compared to other methods (Antunes et al., 2024; Ding et al., 2025).

## H ADDITIONAL CHEMICAL ANALYSIS

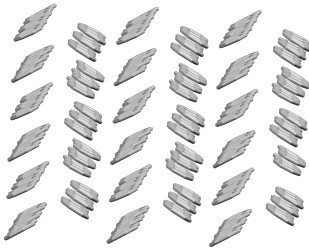

Figure 16: OXTAL learns small local neighborhoods from $S^4$ crops and generalizes to infer large and periodic structures. Example of ANTCEN with over 2400 tokens ($RMSD_{15} = 1.9$Å).

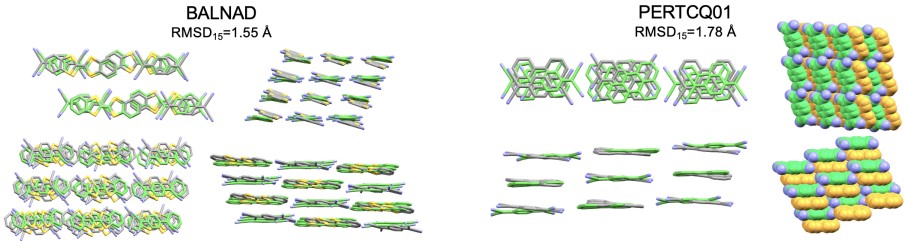

Figure 17: Examples of OXTAL generated co-crystal structures (color) compared against experimental structures (gray).

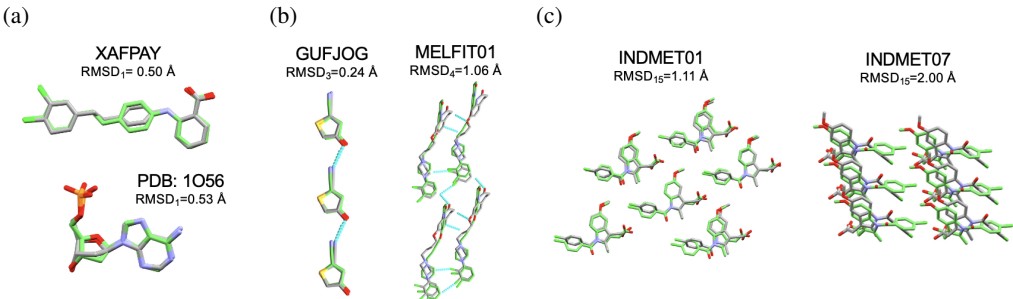

Figure 18: Examples of OXTAL generated structures (color) compared against experimental structures (gray) for (a) flexible conformers, (b) intermolecular interactions, and (c) polymorphs.

### H.1 ENERGY ANALYSIS

To assess whether the generated samples contain (i) highly energetically unfavorable motifs or (ii) chemically plausible packing arrangements compared to physics-based methods, we utilized the

semi-empirical GFN2-xTB method as a computational proxy. GFN2-xTB was selected for its balance of computational speed and accuracy in capturing non-covalent interactions. This analysis should be interpreted as a *consistency check on gross energetic behaviour* rather than as a quantitative ranking of polymorph stability.

We retrieved 1,500 physics-based structure predictions submitted to the CSP blind tests to serve as a baseline. Both the physics-based submissions and the experimental ground truth were expanded into 2×2×2 supercells to match the approximate atom count of the OXTAL generations (30 molecules). For OXTAL samples, hydrogens were added using the CCDC API.

We performed single-point energy (SPE) calculations on all structures. The energy difference is relative to the experimental structure and standardized by number of molecules. It is important to note that the DFT baseline consists of structures that underwent varying degrees of geometry optimization, whereas OXTAL samples were evaluated as raw generative outputs with no geometry optimization, including the hydrogen atoms.

The results show that first, OXTAL samples do not contain energetically catastrophic motifs. Second, despite lacking relaxation, the generative samples exhibit a narrow energy distribution comparable to the stable basin of the DFT samples. This confirms that the generative model implicitly learns to avoid severe steric clashes and produces metastable packing configurations.

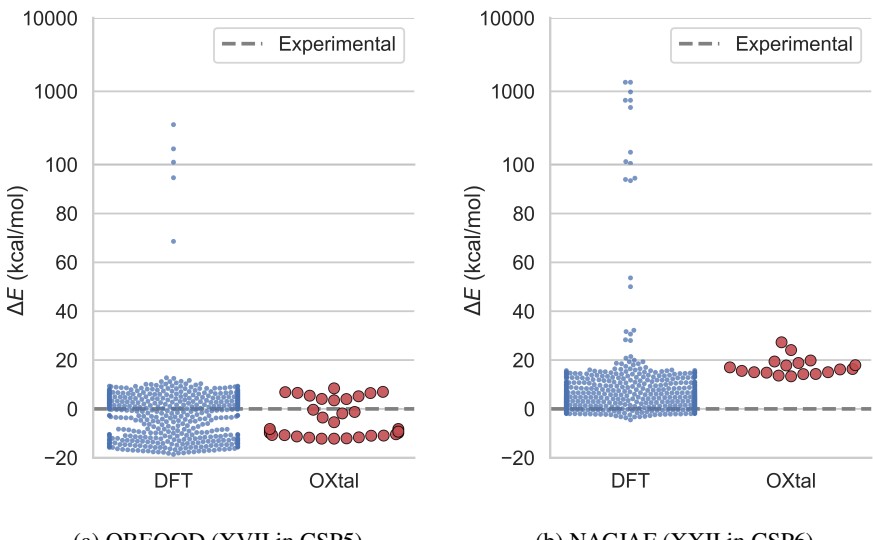

(a) OBEQOD (XVII in CSP5)       (b) NACJAF (XXII in CSP6)

Figure 19: GFN2-xTB single point energy analysis of ground truth crystal structures, 1,500 subsampled physics-based samples from the blind test submissions, and OXTAL samples.

## H.2 Diversity of Generated Samples

In Figure 20, we provide a qualitative visualization of some example packings generated by OXTAL. We see that although several samples lie in the same orientation, some generated samples also present a more complex herringbone packing. This is evident in the XATJOT co-crystal, which does not collapse the two different molecular components into the same orientation. These results suggest that OXTAL may prefer planar packings, which are more prevalent in the training dataset. However, this may be mitigated with additional tuning on noise scheduler parameters.

KONTIQ    OBEQET    OBEQIX    OBEQOD    OBEQUJ    XATJOT

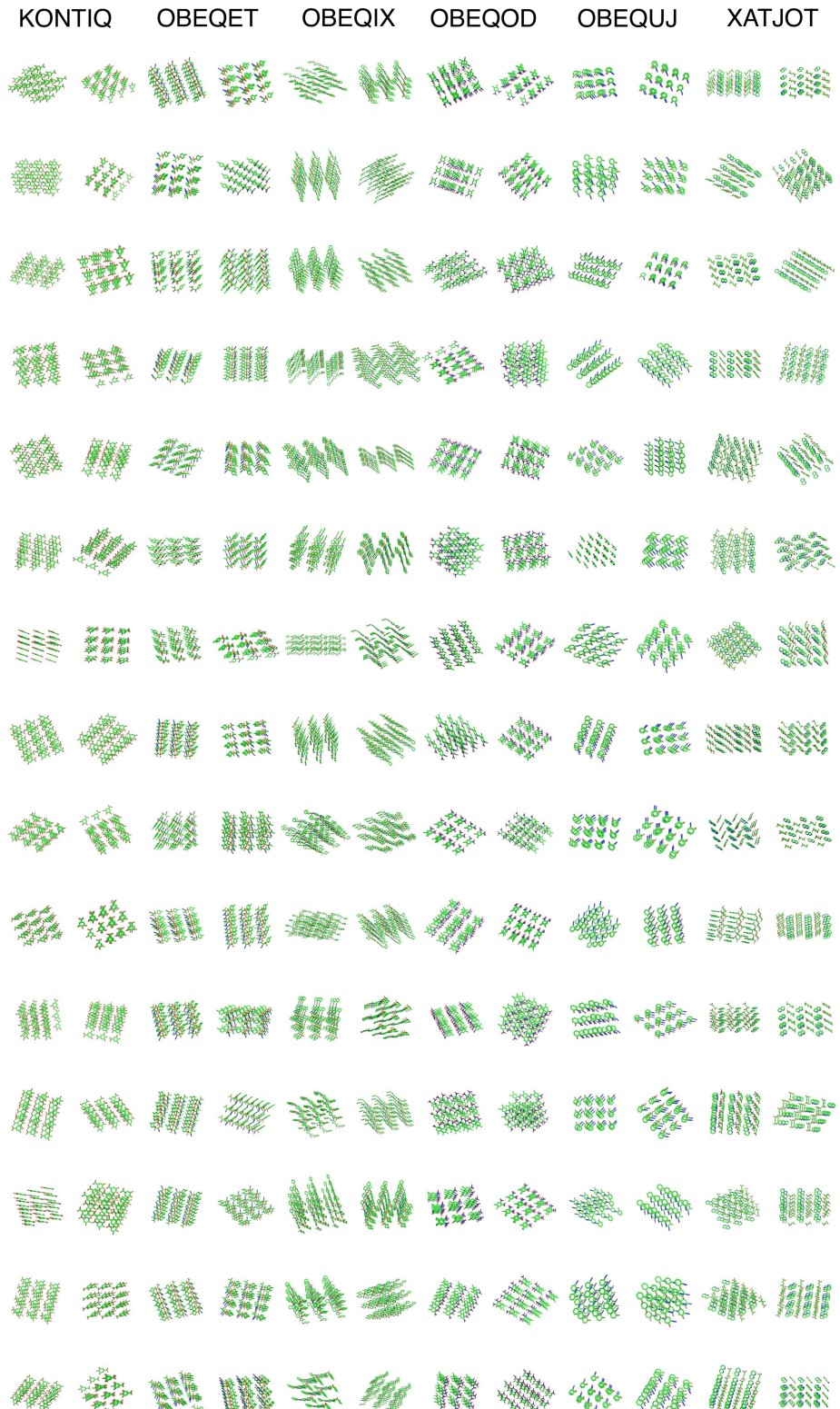

Figure 20: Example set of structures generated by OXTAL for various crystals (labeled by CSD ID).

# I   DETAILED DFT BASELINES FROM CSP BLIND TESTS

Table 8: Results per crystal of submitted methods for CSP blind test 5.

| Model | $n_S$ | $Col_S \downarrow$ | $Pac_S \uparrow$ | $Pac_C$? | $Rec_S \uparrow$ | $Rec_C$? | $\widetilde{Sol}_C$? |
|---|---|---|---|---|---|---|---|
| | | | Target XVI (OBEQUJ) | | | | |
| Group Ammon | 20 | 0.000 | 0.850 | Y | 1.000 | Y | Y |
| Group Boerrigter | 8456 | 0.000 | 0.113 | Y | 0.994 | Y | Y |
| Group Day | 549 | 0.000 | 0.222 | Y | 1.000 | Y | Y |
| Group Desiraju | 202 | 0.000 | 0.272 | Y | 1.000 | Y | Y |
| Group Hofmann | 103 | 0.000 | 0.000 | Y | 1.000 | Y | N |
| Group Jose | 16 | 0.188 | 0.250 | Y | 0.438 | Y | N |
| Group Kendrick | 90 | 0.000 | 0.500 | Y | 1.000 | Y | Y |
| Group Maleev | 18 | 0.000 | 0.000 | N | 0.944 | Y | N |
| Group Misquitta | 103 | 0.000 | 0.214 | Y | 1.000 | Y | Y |
| Group Nikylov | 13 | 0.000 | 0.000 | N | 1.000 | Y | N |
| Group Orendt | 292 | 0.000 | 0.209 | Y | 1.000 | Y | Y |
| Group Price | 150 | 0.000 | 0.367 | Y | 1.000 | Y | Y |
| Group Scheraga | 61 | 0.000 | 0.393 | Y | 1.000 | Y | Y |
| Group vanEijck | 117 | 0.000 | 0.359 | Y | 1.000 | Y | Y |
| Group Venuti | 61 | 0.033 | 0.049 | Y | 0.475 | Y | Y |
| | | | Target XVII (OBEQOD) | | | | |
| Group Ammon | 6 | 0.000 | 0.167 | Y | 1.000 | Y | N |
| Group Boerrigter | 4894 | 0.000 | 0.054 | Y | 0.999 | Y | Y |
| Group Day | 164 | 0.000 | 0.220 | Y | 1.000 | Y | Y |
| Group Desiraju | 202 | 0.000 | 0.302 | Y | 1.000 | Y | Y |
| Group Hofmann | 103 | 0.000 | 0.000 | N | 1.000 | Y | N |
| Group Jose | 15 | 0.533 | 0.133 | Y | 0.267 | Y | N |
| Group Kendrick | 150 | 0.000 | 0.200 | Y | 1.000 | Y | Y |
| Group Maleev | 24 | 0.000 | 0.083 | Y | 1.000 | Y | N |
| Group Orendt | 272 | 0.000 | 0.051 | Y | 0.971 | Y | Y |
| Group Price | 139 | 0.000 | 0.122 | Y | 1.000 | Y | Y |
| Group Scheraga | 74 | 0.000 | 0.189 | Y | 0.986 | Y | Y |
| Group vanEijck | 132 | 0.000 | 0.129 | Y | 0.992 | Y | Y |
| Group Venuti | 71 | 0.563 | 0.014 | Y | 0.211 | Y | N |
| | | | Target XVIII (OBEQET) | | | | |
| Group Ammon | 5 | 0.000 | 0.000 | N | 0.600 | Y | N |
| Group Boerrigter | 5001 | 0.000 | 0.005 | Y | 0.448 | Y | Y |
| Group Day | 438 | 0.000 | 0.000 | N | 0.893 | Y | N |
| Group Desiraju | 156 | 0.000 | 0.038 | Y | 1.000 | Y | Y |
| Group Hofmann | 103 | 0.000 | 0.000 | N | 0.379 | Y | N |
| Group Jose | 11 | 0.364 | 0.000 | N | 0.000 | N | N |
| Group Kendrick | 79 | 0.000 | 0.127 | Y | 0.911 | Y | Y |
| Group Maleev | 10 | 0.000 | 0.000 | N | 0.800 | Y | N |
| Group Orendt | 165 | 0.000 | 0.012 | Y | 0.897 | Y | Y |
| Group Price | 259 | 0.000 | 0.008 | Y | 0.525 | Y | Y |
| Group Scheraga | 71 | 0.000 | 0.014 | Y | 0.986 | Y | Y |
| Group vanEijck | 113 | 0.000 | 0.000 | N | 0.000 | N | N |
| Group Venuti | 100 | 0.000 | 0.000 | N | 0.040 | Y | N |
| | | | Target XIX (XATJOT) | | | | |
| Group Boerrigter | 383 | 0.000 | 0.355 | Y | 1.000 | Y | Y |
| Group Day | 103 | 0.000 | 0.252 | Y | 1.000 | Y | Y |
| Group Desiraju | 202 | 0.000 | 0.262 | Y | 1.000 | Y | Y |
| Group Hofmann | 103 | 0.000 | 0.155 | Y | 1.000 | Y | Y |
| Group Kendrick | 350 | 0.000 | 0.326 | Y | 1.000 | Y | Y |
| Group Maleev | 23 | 0.000 | 0.217 | Y | 1.000 | Y | Y |
| Group Orendt | 279 | 0.000 | 0.140 | Y | 1.000 | Y | Y |
| Group Price | 164 | 0.000 | 0.079 | Y | 1.000 | Y | Y |
| Group Scheraga | 34 | 0.000 | 0.147 | Y | 1.000 | Y | Y |
| Group vanEijck | 129 | 0.000 | 0.457 | Y | 1.000 | Y | Y |
| Group Venuti | 100 | 0.000 | 0.050 | Y | 1.000 | Y | N |
| | | | Target XX (OBEQIX) | | | | |
| Group Ammon | 11 | 0.000 | 0.000 | N | 0.000 | N | N |
| Group Boerrigter | 5005 | 0.000 | 0.000 | N | 0.014 | Y | N |
| Group Day | 53 | 0.000 | 0.094 | Y | 0.358 | Y | Y |
| Group Hofmann | 103 | 0.971 | 0.000 | N | 0.000 | N | N |
| Group Kendrick | 117 | 0.000 | 0.205 | Y | 0.248 | Y | Y |
| Group Maleev | 8 | 0.000 | 0.000 | N | 0.000 | N | N |
| Group Orendt | 156 | 0.000 | 0.006 | Y | 0.006 | Y | Y |
| Group Price | 103 | 0.000 | 0.039 | Y | 0.243 | Y | Y |
| Group vanEijck | 121 | 0.000 | 0.000 | N | 0.017 | Y | N |
| Group Venuti | 100 | 0.020 | 0.000 | N | 0.000 | N | N |
| | | | Target XXI (KONTIQ) | | | | |
| Group Boerrigter | 5002 | 0.000 | 0.902 | Y | 0.914 | Y | Y |
| Group Day | 2350 | 0.000 | 0.964 | Y | 0.966 | Y | Y |
| Group Desiraju | 244 | 0.000 | 0.943 | Y | 0.943 | Y | Y |
| Group Hofmann | 103 | 0.000 | 0.981 | Y | 0.981 | Y | Y |
| Group Kendrick | 1886 | 0.000 | 0.990 | Y | 0.991 | Y | Y |
| Group Maleev | 26 | 0.000 | 0.769 | Y | 0.769 | Y | Y |
| Group Orendt | 594 | 0.000 | 0.928 | Y | 0.941 | Y | Y |
| Group Price | 449 | 0.002 | 0.978 | Y | 0.980 | Y | Y |
| Group vanEijck | 132 | 0.000 | 0.985 | Y | 0.985 | Y | Y |
| Group Venuti | 294 | 0.000 | 0.568 | Y | 0.058 | Y | Y |

Table 9: Results per crystal of submitted methods for CSP blind test 6.

| Model | $n_S$ | $Col_S \downarrow$ | $Pac_S \uparrow$ | $Pac_C?$ | $Rec_S \uparrow$ | $Rec_C?$ | $\overline{Sol}_C?$ |
|---|---|---|---|---|---|---|---|
| | | | Target XXII (NACJAF) | | | | |
| Group01-Chadha-Singh | 100 | 0.000 | 0.010 | Y | 1.000 | Y | N |
| Group02-Cole | 100 | 0.000 | 0.050 | Y | 1.000 | Y | Y |
| Group03-Day | 200 | 0.000 | 0.085 | Y | 1.000 | Y | Y |
| Group04-Dzyabchenko | 100 | 0.000 | 0.170 | Y | 1.000 | Y | Y |
| Group05-vanEijck | 100 | 0.000 | 0.070 | Y | 1.000 | Y | Y |
| Group06-Elking-FustiMolnar | 198 | 0.005 | 0.056 | Y | 0.939 | Y | Y |
| Group07-vandenEnde-Cuppen | 200 | 0.000 | 0.080 | Y | 1.000 | Y | Y |
| Group08-Facelli | 177 | 0.000 | 0.006 | Y | 0.977 | Y | N |
| Group09-Goto-Obata | 200 | 0.000 | 0.060 | Y | 1.000 | Y | Y |
| Group10-Hofmann | 100 | 0.000 | 0.000 | N | 1.000 | Y | N |
| Group11-Lv-Wang-Ma | 200 | 0.000 | 0.000 | N | 1.000 | Y | N |
| Group12-Marom | 600 | 0.028 | 0.032 | Y | 0.757 | Y | Y |
| Group13-Mohamed | 100 | 0.000 | 0.570 | Y | 1.000 | Y | Y |
| Group14-Neumann-Kendrick-Leusen | 100 | 0.000 | 0.070 | Y | 1.000 | Y | Y |
| Group15-Adjiman-Pantelides | 100 | 0.000 | 0.040 | Y | 1.000 | Y | Y |
| Group16-Pickard-et-al | 33 | 0.000 | 0.000 | N | 1.000 | Y | N |
| Group17-Podeszwa | 99 | 0.000 | 0.091 | Y | 1.000 | Y | Y |
| Group18-Price | 200 | 0.000 | 0.045 | Y | 1.000 | Y | Y |
| Group19-Szalewicz-Price | 100 | 0.000 | 0.030 | Y | 1.000 | Y | Y |
| Group20-Tuckerman-Szalewicz | 54 | 0.000 | 0.130 | Y | 1.000 | Y | Y |
| Group21-Zhu | 80 | 0.000 | 0.075 | Y | 0.988 | Y | Y |
| Group22-Boese-Hofmann | 31 | 0.000 | 0.000 | N | 1.000 | Y | N |
| Group23-Brandenburg-Grimme | 119 | 0.000 | 0.042 | Y | 0.992 | Y | Y |
| Group25-Tkatchenko | 120 | 0.000 | 0.067 | Y | 0.992 | Y | Y |
| | | | Target XXIII (XAFPAY) | | | | |
| Group01-Chadha-Singh | 100 | 0.000 | 0.000 | N | 0.000 | N | N |
| Group02-Cole | 100 | 0.000 | 0.050 | Y | 0.090 | Y | Y |
| Group03-Day | 200 | 0.000 | 0.040 | Y | 0.035 | Y | Y |
| Group05-vanEijck | 100 | 0.000 | 0.040 | Y | 0.060 | Y | Y |
| Group06-Elking-FustiMolnar | 149 | 0.000 | 0.114 | Y | 0.094 | Y | Y |
| Group07-vandenEnde-Cuppen | 200 | 0.000 | 0.000 | N | 0.000 | N | N |
| Group08-Facelli | 100 | 0.000 | 0.000 | N | 0.000 | N | N |
| Group09-Goto-Obata | 200 | 0.000 | 0.120 | Y | 0.135 | Y | Y |
| Group10-Hofmann | 100 | 0.000 | 0.000 | N | 0.000 | N | N |
| Group13-Mohamed | 100 | 0.000 | 0.090 | Y | 0.350 | Y | Y |
| Group14-Neumann-Kendrick-Leusen | 200 | 0.000 | 0.170 | Y | 0.270 | Y | Y |
| Group15-Adjiman-Pantelides | 100 | 0.000 | 0.100 | Y | 0.120 | Y | Y |
| Group18-Price | 200 | 0.000 | 0.120 | Y | 0.165 | Y | Y |
| Group21-Zhu | 60 | 0.000 | 0.083 | Y | 0.133 | Y | Y |
| Group22-Boese-Hofmann | 18 | 0.000 | 0.000 | N | 0.000 | N | N |
| Group23-Brandenburg-Grimme | 125 | 0.000 | 0.400 | Y | 0.304 | Y | Y |
| Group25-Tkatchenko | 50 | 0.000 | 0.280 | Y | 0.220 | Y | Y |
| | | | Target XXIV (XAFQON) | | | | |
| Group03-Day | 200 | 0.980 | 1.000 | Y | 0.020 | Y | Y |
| Group05-vanEijck | 100 | 0.000 | 0.980 | Y | 0.980 | Y | Y |
| Group06-Elking-FustiMolnar | 198 | 0.000 | 0.859 | Y | 0.005 | Y | Y |
| Group08-Facelli | 100 | 0.280 | 0.990 | Y | 0.720 | Y | Y |
| Group10-Hofmann | 100 | 0.000 | 0.990 | Y | 1.000 | Y | Y |
| Group14-Neumann-Kendrick-Leusen | 100 | 0.310 | 1.000 | Y | 0.690 | Y | Y |
| Group18-Price | 200 | 0.805 | 0.990 | Y | 0.195 | Y | Y |
| Group21-Zhu | 50 | 0.140 | 1.000 | Y | 0.860 | Y | Y |
| Group22-Boese-Hofmann | 15 | 0.200 | 1.000 | Y | 0.800 | Y | Y |
| Group23-Brandenburg-Grimme | 119 | 0.941 | 0.966 | Y | 0.059 | Y | Y |
| Group24-Szalewicz | 100 | 0.800 | 1.000 | Y | 0.200 | Y | Y |
| Group25-Tkatchenko | 50 | 0.600 | 1.000 | Y | 0.400 | Y | Y |
| | | | Target XXV (XAFQAZ01) | | | | |
| Group02-Cole | 100 | 0.000 | 0.000 | N | 1.000 | Y | N |
| Group03-Day | 200 | 0.000 | 0.100 | Y | 1.000 | Y | Y |
| Group04-Dzyabchenko | 161 | 0.000 | 0.056 | Y | 1.000 | Y | Y |
| Group05-vanEijck | 100 | 0.000 | 0.060 | Y | 1.000 | Y | Y |
| Group06-Elking-FustiMolnar | 127 | 0.000 | 0.087 | Y | 0.984 | Y | Y |
| Group07-vandenEnde-Cuppen | 200 | 0.000 | 0.000 | N | 0.990 | Y | N |
| Group08-Facelli | 100 | 0.000 | 0.040 | Y | 1.000 | Y | N |
| Group09-Goto-Obata | 200 | 0.000 | 0.030 | Y | 1.000 | Y | N |
| Group10-Hofmann | 100 | 0.000 | 0.000 | N | 1.000 | Y | N |
| Group13-Mohamed | 100 | 0.000 | 0.020 | Y | 1.000 | Y | Y |
| Group14-Neumann-Kendrick-Leusen | 100 | 0.000 | 0.100 | Y | 1.000 | Y | Y |
| Group15-Adjiman-Pantelides | 100 | 0.000 | 0.090 | Y | 1.000 | Y | Y |
| Group18-Price | 200 | 0.000 | 0.110 | Y | 1.000 | Y | Y |
| Group21-Zhu | 25 | 0.000 | 0.040 | Y | 1.000 | Y | Y |
| Group22-Boese-Hofmann | 10 | 0.000 | 0.000 | N | 1.000 | Y | N |
| Group23-Brandenburg-Grimme | 100 | 0.000 | 0.110 | Y | 1.000 | Y | Y |
| Group25-Tkatchenko | 20 | 0.000 | 0.100 | Y | 1.000 | Y | Y |
| | | | Target XXVI (XAFQIH) | | | | |
| Group01-Chadha-Singh | 100 | 0.000 | 0.000 | N | 0.000 | N | N |
| Group02-Cole | 100 | 0.000 | 0.010 | Y | 0.010 | Y | Y |
| Group03-Day | 200 | 0.000 | 0.015 | Y | 0.000 | N | Y |
| Group04-Dzyabchenko | 71 | 0.000 | 0.014 | Y | 0.000 | N | Y |
| Group05-vanEijck | 100 | 0.000 | 0.000 | N | 0.000 | N | N |
| Group06-Elking-FustiMolnar | 138 | 0.000 | 0.312 | Y | 0.246 | Y | Y |
| Group10-Hofmann | 100 | 0.000 | 0.000 | N | 0.000 | N | N |
| Group13-Mohamed | 100 | 0.000 | 0.000 | N | 0.000 | N | N |
| Group14-Neumann-Kendrick-Leusen | 200 | 0.000 | 0.405 | Y | 0.490 | Y | Y |
| Group15-Adjiman-Pantelides | 100 | 0.000 | 0.020 | Y | 0.000 | N | Y |
| Group18-Price | 200 | 0.000 | 0.035 | Y | 0.040 | Y | Y |
| Group21-Zhu | 30 | 0.000 | 0.000 | N | 0.000 | N | N |
| Group22-Boese-Hofmann | 15 | 0.000 | 0.000 | N | 0.000 | N | N |
| Group23-Brandenburg-Grimme | 103 | 0.000 | 0.019 | Y | 0.058 | Y | Y |

Table 10: Results per crystal of submitted methods for CSP blind test 7.

| Model | $n_S$ | $Col_S$ ↓ | $Pac_S$ ↑ | $Pac_C$ ↑ | $Rec_S$ ↑ | $Rec_C$ ↑ | $\overline{Sol}_C$ ↑ |
|---|---|---|---|---|---|---|---|
| **Target XXVII (XIFZOF01)** | | | | | | | |
| Group06-BEijck | 1500 | 0.000 | 0.011 | Y | 0.006 | Y | Y |
| Group08-DWMHofmann | 1501 | 0.000 | 0.001 | Y | 0.000 | N | N |
| Group10-XtalPi | 1500 | 0.000 | 0.062 | Y | 0.038 | Y | Y |
| Group16-Marom-Isayev | 1500 | 0.000 | 0.027 | Y | 0.005 | Y | Y |
| Group17-Matsui | 1500 | 0.000 | 0.005 | Y | 0.005 | Y | Y |
| Group19-OpenEye | 1500 | 0.000 | 0.000 | N | 0.000 | N | N |
| Group20-MNeumann | 1500 | 0.000 | 0.015 | Y | 0.007 | Y | Y |
| Group21-Obata-Goto | 1500 | 0.000 | 0.011 | Y | 0.003 | Y | Y |
| Group22-AOganov | 1500 | 0.000 | 0.014 | Y | 0.000 | N | Y |
| Group24-SLPrice | 1500 | 0.000 | 0.005 | Y | 0.000 | N | Y |
| Group25-CShang | 1500 | 0.000 | 0.061 | Y | 0.007 | Y | Y |
| Group27-KSzalewicz-MTuckerman | 1500 | 0.000 | 0.027 | Y | 0.000 | N | Y |
| Group28-QZhu | 1500 | 0.000 | 0.000 | N | 0.000 | N | N |
| **Target XXVIII (OJIGOG01)** | | | | | | | |
| Group06-BEijck | 1500 | 0.076 | 0.002 | Y | 0.000 | N | Y |
| Group08-DWMHofmann | 31 | 1.000 | 0.516 | Y | 0.000 | N | N |
| Group10-XtalPi | 1500 | 1.000 | 0.141 | Y | 0.000 | N | N |
| Group20-MNeumann | 1500 | 0.947 | 0.127 | Y | 0.008 | Y | N |
| Group22-AOganov | 811 | 0.335 | 0.002 | Y | 0.000 | N | N |
| Group24-SLPrice | 1500 | 1.000 | 0.093 | Y | 0.000 | N | N |
| Group25-CShang | 1500 | 0.993 | 0.359 | Y | 0.005 | Y | Y |
| Group26-KSzalewicz | 1500 | 1.000 | 0.000 | N | 0.000 | N | N |
| **Target XXIX (FASMEV)** | | | | | | | |
| Group01-Adjiman-Pantelides | 1510 | 0.000 | 0.072 | Y | 0.961 | Y | Y |
| Group03-DBoese | 1510 | 0.000 | 0.060 | Y | 0.836 | Y | Y |
| Group05-GDay | 1510 | 0.000 | 0.143 | Y | 0.981 | Y | Y |
| Group06-BEijck | 1510 | 0.000 | 0.052 | Y | 0.668 | Y | Y |
| Group10-XtalPi | 1510 | 0.000 | 0.223 | Y | 0.997 | Y | Y |
| Group11-Johnson-Otero-de-la-Roza | 319 | 0.000 | 0.245 | Y | 0.997 | Y | Y |
| Group12-JJose | 2096 | 0.000 | 0.147 | Y | 0.000 | N | N |
| Group13-DKhakimov | 80 | 0.000 | 0.537 | Y | 1.000 | Y | Y |
| Group16-Marom-Isayev | 1510 | 0.000 | 0.476 | Y | 1.000 | Y | Y |
| Group18-SMohamed | 1510 | 0.000 | 0.054 | Y | 0.984 | Y | Y |
| Group19-OpenEye | 904 | 0.000 | 0.043 | Y | 0.893 | Y | Y |
| Group20-MNeumann | 1504 | 0.000 | 0.418 | Y | 0.999 | Y | Y |
| Group21-Obata-Goto | 1510 | 0.000 | 0.079 | Y | 0.738 | Y | Y |
| Group22-AOganov | 1510 | 0.000 | 0.054 | Y | 0.719 | Y | Y |
| Group23-CJPickard | 1510 | 0.001 | 0.154 | Y | 0.994 | Y | Y |
| Group24-SLPrice | 1265 | 0.000 | 0.043 | Y | 0.999 | Y | Y |
| Group25-CShang | 1500 | 0.000 | 0.159 | Y | 0.988 | Y | Y |
| Group27-KSzalewicz-MTuckerman | 1510 | 0.000 | 0.175 | Y | 0.972 | Y | Y |
| Group28-QZhu | 209 | 0.000 | 0.096 | Y | 0.947 | Y | Y |
| **Target XXX - 2:1 (MIVZEA)** | | | | | | | |
| Group05-GDay | 1600 | 0.000 | 0.007 | Y | 0.379 | Y | Y |
| Group06-BEijck | 1600 | 0.000 | 0.007 | Y | 0.239 | Y | Y |
| Group10-XtalPi | 1600 | 0.000 | 0.042 | Y | 0.186 | Y | Y |
| Group12-JJose | 403 | 0.000 | 0.000 | N | 0.432 | Y | N |
| Group13-DKhakimov | 281 | 0.000 | 0.000 | N | 1.000 | Y | N |
| Group18-SMohamed | 1600 | 0.000 | 0.002 | Y | 1.000 | Y | Y |
| Group19-OpenEye | 1600 | 0.000 | 0.001 | Y | 0.052 | Y | Y |
| Group20-MNeumann | 1600 | 0.000 | 0.042 | Y | 0.207 | Y | Y |
| Group21-Obata-Goto | 1600 | 0.000 | 0.002 | Y | 0.541 | Y | Y |
| Group22-AOganov | 1138 | 0.000 | 0.000 | N | 0.149 | Y | N |
| Group24-SLPrice | 1600 | 0.000 | 0.000 | N | 0.269 | Y | N |
| Group27-KSzalewicz-MTuckerman | 1600 | 0.000 | 0.001 | Y | 0.287 | Y | Y |
| Group28-QZhu | 1600 | 0.000 | 0.001 | Y | 0.070 | Y | Y |
| **Target XXX - 1:1 (MIVZIE)** | | | | | | | |
| Group05-GDay | 1600 | 0.000 | 0.016 | Y | 0.099 | Y | Y |
| Group06-BEijck | 1600 | 0.000 | 0.004 | Y | 0.041 | Y | Y |
| Group10-XtalPi | 1600 | 0.000 | 0.024 | Y | 0.107 | Y | Y |
| Group12-JJose | 403 | 0.000 | 0.000 | N | 0.000 | N | N |
| Group13-DKhakimov | 281 | 0.000 | 0.007 | Y | 0.000 | N | N |
| Group18-SMohamed | 1600 | 0.000 | 0.000 | N | 0.000 | N | N |
| Group19-OpenEye | 1600 | 0.000 | 0.001 | Y | 0.046 | Y | N |
| Group20-MNeumann | 1600 | 0.000 | 0.015 | Y | 0.111 | Y | Y |
| Group21-Obata-Goto | 1600 | 0.000 | 0.006 | Y | 0.001 | Y | Y |
| Group22-AOganov | 1138 | 0.000 | 0.001 | Y | 0.000 | N | N |
| Group24-SLPrice | 1600 | 0.000 | 0.001 | Y | 0.000 | N | N |
| Group27-KSzalewicz-MTuckerman | 1600 | 0.000 | 0.001 | Y | 0.016 | Y | N |
| Group28-QZhu | 1600 | 0.000 | 0.003 | Y | 0.083 | Y | Y |
| **Target XXXI (ZEHFUR)** | | | | | | | |
| Group01-Adjiman-Pantelides | 1500 | 0.000 | 0.029 | Y | 0.330 | Y | Y |
| Group03-DBoese | 1500 | 0.000 | 0.029 | Y | 0.266 | Y | Y |
| Group05-GDay | 1500 | 0.005 | 0.055 | Y | 0.351 | Y | Y |
| Group06-BEijck | 1500 | 0.000 | 0.008 | Y | 0.029 | Y | Y |
| Group08-DWMHofmann | 1500 | 0.000 | 0.004 | Y | 0.199 | Y | Y |
| Group10-XtalPi | 1500 | 0.000 | 0.055 | Y | 0.513 | Y | Y |
| Group12-JJose | 1500 | 0.001 | 0.025 | Y | 0.000 | N | N |
| Group16-Marom-Isayev | 1500 | 0.000 | 0.063 | Y | 0.405 | Y | Y |
| Group18-SMohamed | 1500 | 0.000 | 0.001 | Y | 0.000 | N | N |
| Group19-OpenEye | 1500 | 0.000 | 0.030 | Y | 0.090 | Y | Y |
| Group20-MNeumann | 1500 | 0.000 | 0.123 | Y | 0.636 | Y | Y |
| Group21-Obata-Goto | 1500 | 0.000 | 0.006 | Y | 0.137 | Y | Y |
| Group22-AOganov | 1500 | 0.000 | 0.003 | Y | 0.098 | Y | Y |
| Group24-SLPrice | 1500 | 0.000 | 0.017 | Y | 0.244 | Y | Y |
| Group25-CShang | 1500 | 0.000 | 0.069 | Y | 0.678 | Y | Y |
| Group27-KSzalewicz-MTuckerman | 1500 | 0.000 | 0.009 | Y | 0.073 | Y | Y |
| Group28-QZhu | 1500 | 0.000 | 0.001 | Y | 0.045 | Y | N |
| **Target XXXII (JEKVII)** | | | | | | | |
| Group01-Adjiman-Pantelides | 1499 | 0.006 | 0.001 | Y | 0.000 | N | N |
| Group03-DBoese | 1500 | 0.000 | 0.000 | N | 0.000 | N | N |
| Group05-GDay | 1500 | 0.000 | 0.000 | N | 0.000 | N | N |
| Group06-BEijck | 1500 | 0.000 | 0.000 | N | 0.000 | N | N |
| Group10-XtalPi | 1500 | 0.001 | 0.030 | Y | 0.020 | Y | Y |
| Group18-SMohamed | 203 | 0.000 | 0.000 | N | 0.000 | N | N |
| Group19-OpenEye | 1500 | 0.000 | 0.000 | N | 0.000 | N | N |
| Group20-MNeumann | 1500 | 0.000 | 0.134 | Y | 0.066 | Y | Y |
| Group22-AOganov | 1500 | 0.000 | 0.000 | N | 0.000 | N | N |
| Group24-SLPrice | 1500 | 0.000 | 0.000 | N | 0.000 | N | N |
| Group25-CShang | 1500 | 0.000 | 0.085 | Y | 0.014 | Y | Y |
| Group27-KSzalewicz-MTuckerman | 1500 | 0.000 | 0.000 | N | 0.000 | N | N |
| Group28-QZhu | 1495 | 0.000 | 0.000 | N | 0.000 | N | N |
| **Target XXXIII (ZEGWAN)** | | | | | | | |
| Group01-Adjiman-Pantelides | 1500 | 0.000 | 0.011 | Y | 0.429 | Y | Y |
| Group05-GDay | 1500 | 0.000 | 0.013 | Y | 0.623 | Y | Y |
| Group06-BEijck | 1500 | 0.000 | 0.015 | Y | 0.338 | Y | Y |
| Group08-DWMHofmann | 1500 | 0.000 | 0.004 | Y | 0.522 | Y | Y |
| Group10-XtalPi | 1500 | 0.000 | 0.029 | Y | 0.797 | Y | Y |
| Group13-DKhakimov | 56 | 0.000 | 0.000 | N | 0.000 | N | N |
| Group19-OpenEye | 1500 | 0.000 | 0.022 | Y | 0.758 | Y | Y |
| Group20-MNeumann | 1500 | 0.000 | 0.052 | Y | 0.729 | Y | Y |
| Group21-Obata-Goto | 1500 | 0.000 | 0.010 | Y | 0.553 | Y | Y |
| Group22-AOganov | 1500 | 0.000 | 0.000 | N | 0.008 | Y | N |
| Group24-SLPrice | 1500 | 0.000 | 0.013 | Y | 0.363 | Y | Y |
| Group25-CShang | 1500 | 0.000 | 0.009 | Y | 0.545 | Y | Y |
| Group27-KSzalewicz-MTuckerman | 1500 | 0.000 | 0.009 | Y | 0.231 | Y | Y |
| Group28-QZhu | 1453 | 0.000 | 0.012 | Y | 0.551 | Y | Y |

Table 11: Summary of computational resources used by some of the participants as reported in the 5th CSP blind test. CPU hours are approximately normalized to 3.0 GHz. OXTAL times are reported as elapsed wall time in hours on 1 L40S GPU and 6 CPUs.

| Group | XVI | XVII | XVIII | XIX | XX | XXI | Total |
|---|---|---|---|---|---|---|---|
| Boerrigter | 90 | 100 | 350 | 650 | 2,105 | 600 | 3,800 |
| Day, Cruz-Cabeza | 110 | 1,941 | 21,051 | 6,097 | 54,090 | 22,197 | 91,400 |
| Desiraju, Thakur, Tiwari, Pal | 114 | 2,303 | 324 | 114 | | 1,431 | 4,600 |
| Hofmann | 2 | 7 | 12 | 694 | 670 | 187 | 1,600 |
| Neumann, Leusen, Kendrick, van de Streek | | | | | | | 115,000 |
| Price et al. | 200 | 5,000 | 14,000 | 3,000 | 120,000 | 52,800 | 195,000 |
| Van Eijck | 27 | | | | | | 9,500 |
| Della Valle, Venuti | | | | | | | 3,200 |
| Maleev, Zhitkov | | | | | | | 7,500 |
| Misquitta, Pickard & Needs | | | | | | | 162,000 |
| Scheraga, Arnautova | 150 | 150 | 610 | 720 | | | 1,300 |
| OXTAL | 0.024 | 0.011 | 0.012 | 0.013 | 0.029 | 0.011 | 0.100 |

Table 12: Summary of the computational resources used by each submission in terms of raw CPU hours as reported in the 6th CSP blind test. OXTAL times are reported as elapsed wall time in hours on 1 L40S GPU and 6 CPUs. Note that Group 22 re-ranks structures submitted by Group 10, and Groups 23, 24, and 25 re-rank structures submitted by Group 18.

| Group | XXII | XXIII | XXIV | XXV | XXVI | Total |
|---|---|---|---|---|---|---|
| 01-Chadha & Singh | 350 | 450 | | | 600 | 1,400 |
| 02-Cole et al. | 6 | 538 | | 46 | 246 | 836 |
| 03-Day et al. | 12,714 | 394,948 | 15,241 | 121,701 | 179,897 | 724,501 |
| 04-Dzyabchenko | 144 | 3,648 | 3,360 | | | 7,152 |
| 05-van Eijck | 130 | 2,810 | 1,400 | 8,060 | 7,630 | 20,030 |
| 06-Elking & Fusti-Molnar | 418,540 | 242,000 | 235,400 | 135,000 | 190,000 | 1,220,940 |
| 07-van den Ende, Cuppen et al. | 9,741 | 7,777 | | 6,388 | 23,906 | |
| 08-Facelli et al. | 268,012 | 38,500 | 11,500 | 39,000 | | 357,012 |
| 09-Obata & Goto | 19,200 | 346,000 | | 325,000 | | 690,200 |
| 10-Hofmann & Kuleshova | 10 | 630 | 623 | 202 | 255 | 1,720 |
| 11-Lv, Wang, Ma | 325,000 | | | | | 325,000 |
| 12-Marom et al. | 30,000,000 | | | | | 30,000,000 |
| 13-Mohamed | 26 | 106 | | 81 | 61 | 274 |
| 14-Neumann, Kendrick, Leusen | 32,160 | 146,120 | 103,700 | 84,680 | 356,844 | 723,504 |
| 15-Pantelides, Adjiman et al. | 333 | 87,000 | | 37,535 | 272,500 | 397,368 |
| 16-Pickard et al. | 380,000 | | | | | 380,000 |
| 17-Podeszwa et al. | 72,220 | | | | | 72,220 |
| 18-Price et al. | 26,000 | 84,000 | 63,000 | 169,000 | 327,000 | 669,000 |
| 19-Szalewicz et al. | 66,000 | | | | | 66,000 |
| 20-Tuckerman, Szalewicz et al. | 81,000 | | | | | 81,000 |
| 21-Zhu, Oganov, Masunov | 4,000 | 275,000 | 279,800 | 30,000 | 180,000 | 768,800 |
| 22-Boese | 80,000 | 80,000 | 80,000 | 80,000 | 80,000 | 400,000 |
| 23-Brandenburg & Grimme | 13,665 | 8,661 | 3,509 | 34,824 | 10,135 | 70,794 |
| 24-Szalewicz et al. | | | 15,000 | | | 15,000 |
| 25-Tkatchenko et al. | 100,000 | 2,100,000 | 500,000 | 500,000 | | 3,200,000 |
| OXTAL | 0.012 | 0.019 | 0.011 | 0.030 | 0.042 | 0.114 |

Table 13: CPU core hours per target molecule for each prediction method as reported in the 7th CSP blind test. OXTAL times are reported as elapsed wall time in hours on 1 L40S GPU and 6 CPUs.

| Group | XXVII | XXVIII | XXIX | XXX | XXXI | XXXII | XXXIII | Total |
|---|---|---|---|---|---|---|---|---|
| 01-Adjiman-Pantelides | | | 652,495 | | 840,000 | 1,597,000 | 412,000 | 3,501,495 |
| 03-DBoese | | | 1,600,000 | | 1,500,000 | 3,600,000 | | 6,700,000 |
| 05-GDay | 768,766 | | 33,000 | 2,900,000 | 510,563 | 846,698 | 228,957 | 5,287,984 |
| 06-BEijck | 8,120 | 1,350 | 1,310 | 9,800 | 1,470 | 2,900 | 4,980 | 29,930 |
| 08-DWMHofmann | 3,200 | 10 | | | 4,000 | | 1,840 | 9,050 |
| 10-XtalPi | 772,500 | 1,242,500 | 1,146,588 | 644,927 | 381,672 | 644,927 | 612,500 | 5,445,614 |
| 11-Johnson-Otero-de-la-Roza | | | 643,882 | | | | | 643,882 |
| 12-JJose | | | 20,000 | 80,000 | 20,000 | | | 120,000 |
| 13-DKhakimov | | | 350 | 1,500 | | | 500 | 2,350 |
| 16-Marom-Isayev | 1,700,000 | | 2,128,000 | | 630,000 | | | 4,458,000 |
| 17-Matsui | 95,819 | | | | | | | 95,819 |
| 18-SMohamed | | | 1,050 | 36,864 | 632 | 1,561 | | 40,107 |
| 19-OpenEye | 30,000 | | 40,000 | 1,250,000 | 140,000 | 400,000 | 60,000 | 1,920,000 |
| 20-MNeumann | 1,022,976 | 283,538 | 755,712 | 1,769,472 | 1,028,064 | 3,935,232 | 728,064 | 9,523,058 |
| 21-Obata-Goto | 333,586 | | 92,890 | 580,436 | 1,889,649 | | 477,210 | 3,373,771 |
| 22-AOganov | 20,000 | 2,000 | 15,000 | 180,000 | 20,000 | 25,000 | 25,000 | 287,000 |
| 23-CJPickard | | | 10,000 | | | | | 10,000 |
| 24-SLPrice | 450,290 | 89,666 | 76,541 | 100,000 | 49,177 | 244,520 | 123,427 | 1,133,621 |
| 25-CShang | 55,150 | 29,691 | 4,784 | | 6,476 | 76,161 | 34,648 | 206,910 |
| 26-KSzalewicz | | | | | 28,332 | | | 28,332 |
| 27-KSzalewicz-MTuckerman | 1,280,566 | 60,457 | 242,424 | 213,722 | | 1,663,940 | 150,650 | 3,611,759 |
| 28-QZhu | 1,600 | | 1,500 | 7,680 | 1,500 | 1,500 | 1,500 | 15,280 |
| OXTAL | 0.060 | 0.026 | 0.011 | 0.056 | 0.015 | 0.050 | 0.017 | 0.235 |

