# OpenReview forum: "OXtal: An All-Atom Diffusion Model for Organic Crystal Structure Prediction"
_ICLR.cc/2026/Conference — ICLR 2026 Poster_

### Official Review · Reviewer_cVf8 · 2025-10-27

**Soundness:** 3
**Presentation:** 3
**Contribution:** 3
**Rating:** 8
**Confidence:** 4

**Summary:**

The paper introduces OXTAL, an all-atom diffusion model for molecular crystal structure prediction (CSP).
Trained on ∼600k crystals from the Cambridge Structural Database (CSD), OXTAL learns the joint distribution of molecular conformations and periodic packings using a new S4 (Stoichiometric Stochastic Shell Sampling) scheme.
It achieves strong performance on CCDC blind tests with competitive lattice recovery compared to DFT-based CSP at vastly lower cost.

**Strengths:**

This is among the most technically compelling works to date on molecular crystal structure prediction, and, to the best of my knowledge, the first diffusion-based framework that addresses molecular crystals comprehensively.
- Innovative lattice-free diffusion formulation (S4).
- Extensive benchmarks and realistic evaluation metrics (RMSD₁, RMSD₁₅, Smooth-LDDT).
- Strong empirical results and clear scientific motivation.

**Weaknesses:**

- The training dataset (CSD) is under commercial license, and the authors will not release the training code. This prevents full replication of results.
- The reason given (“CSD redistribution restrictions”) does not strictly apply to code; code could be released with dummy data-loading interfaces.
- The paper does not evaluate OXTAL on public molecular crystal datasets (e.g., COD, MP-organic subsets), so it is unclear whether the method generalizes beyond CSD.
- The paper could include more quantitative analysis on the impact of shell radius or token size in S4 on model accuracy.
- Minor language and formatting issues:
  * “do we 2 rounds of recycling” → should be “we do 2 rounds of recycling.”
  * “orders-of-magnitude lower inference cost” → better phrased as “several orders of magnitude lower.”
  * Variables  L and  d in Eqs. (21–27) are undefined in the main text.

**Questions:**

- Could the authors clarify why the training code cannot be released if it only depends on the data interface, not the CSD files themselves?
- How does OXTAL perform on public datasets (e.g., COD) or inorganic datasets to demonstrate cross-domain generalization?
- What is the impact of using non-equivariant Transformer architecture compared to SE(3)-equivariant baselines such as GemNet-OC or EquiformerV2 in this setting?
- Could a smaller-scale variant of OXTAL (e.g., <100 M parameters) achieve comparable results with fewer GPU resources?

---

> ### Author Response · Authors · 2025-11-24
>
> We sincerely thank Reviewer cVf8 for their review and positive appraisal of our work. We are delighted to hear that the reviewer considers this "one of the most technically compelling" and innovative works on molecular CSP. We also appreciate that the reviewer finds our empirical evaluation to contain extensive benchmarks, realistic evaluation metrics, and strong empirical performance. We believe these are essential to making true progress in this kind of interdisciplinary research. We now address the main clarification points raised in the review.
>
> **Release of Training Code**
>
> We acknowledge the reviewer's request for the release of our training code. We would like to clarify that we do, in fact, intend to release our training code as we firmly believe in open and reproducible science. Our initial submission erred on the side of caution, and we are currently in the process of preparing a code release that conforms to the CCDC licence and would allow for the full replication of our results. We anticipate this to occur imminently after the conclusion of this review process. We hope this now assuages the reviewers' concern.
>
> **Evaluation on Public Datasets (e.g. COD)**
>
> We appreciate the reviewer’s comment on the need to evaluate OXtal on the publicly available COD dataset. We first clarify that the COD is mostly a subset of the comprehensive CSD dataset that we use in this work, and thus these two datasets are closely analogous. For example, rubrene in our evaluation set is in COD (ID 1516682). Additionally, if by ‘MP-organic subsets’ the reviewer meant the MOF subset of Materials Project, the experimental data are derived from CSD as well. Furthermore, we note that inorganic crystals provide a fundamentally different underlying data distribution since they do not contain molecular conformers, are much smaller, and are characterized by strong covalent or ionic bonds (as opposed to the weak, longer ranged bonds in molecular crystals).
>
> **Additional Ablation Studies**
>
> We thank the reviewer for their suggestions on additional ablations, which have strengthened our work.
>
> 1. **$S^4$ Shell Radius**
>
> From our analysis on $S^4$ shell radius in our global response, we see that the impact of radius size is minimal for reasonable choices of radius, since all three settings still outperform existing cropping methods. Furthermore, we see that there is indeed some interplay between shell radius size and token size as evidenced by the results from using a larger 5.5 shell radius size, which limits the number of full shells that fit within the total token budget. We thank the reviewer for this insightful suggestion and have included a discussion of this analysis in Appendix D.2 of our updated PDF.
>
> 2. **Equivariance Ablation**
>
> The reviewer raises a great point stating:
>
> >What is the impact of using non-equivariant Transformer architecture compared to SE(3)-equivariant baselines such as GemNet-OC or EquiformerV2 in this setting?
>
> We have provided an equivariance ablation study with EquiformerV2 in our global response. These results show that using a non-equivariant model is essential to being able to scale OXTal to the size of larger molecular crystals.
>
> 3. **Model Size Ablation**
>
> >Could a smaller-scale variant of OXTAL (e.g., <100 M parameters) achieve comparable results with fewer GPU resources?
>
> We provide an ablation here with a smaller model that is ~50M parameters. As usual, these models were trained for 50k steps, and results are averaged across all 5 datasets, with more detailed results reported in Appendix D.4 of our updated pdf.
>
> | $\text{Total Params}$   | $Col_s$ ↓	| $Lat_S$ ↑	| $Lat_C$ ↑  	| $Rec_S$ ↑  	| $Rec_C$ ↑  	| $\widetilde{Sol}_C$ ↑ 	|
> |--------|------------|------------|--------------|--------------|--------------|------------|
> | A-Transformer	| 0.8710     | 0.0034 	| 0.0246   	| 0.0066   	| 0.0240   	| 0.0120 	|
> | AssembleFlow	| 0.7398     | 0.0002 	| 0.008   	| 0.0574   	| 0.1940   	| 0 |
> | OXtal (50M)	| 0.1268     | 0.2220 	| 0.6584   	| 0.1188   	| 0.4702   	| 0.1094 	|
> | OXtal (100M)  | **0.0816** | **0.4390** | **0.8296**  | **0.2736**   | **0.7356**   | **0.1336** |
>
> The performance of OXtal decreases quite significantly with a smaller model, suggesting that being able to successfully scale the model is one of the major keys of success. That said, even the smaller 50M parameter version of OXtal is still able to outperform existing _ab initio_ ML baselines.
>
> **Formatting Issues**
>
> We thank the reviewer for catching the formatting and presentation issues in our manuscript. We have corrected them in our updated manuscript.
>
> **Concluding Remarks**
>
> We thank the reviewer for their time and dedication to the review process. We hope that our answers here have fully answered all the great questions raised by the reviewer and allows them to continue endorsing our work strongly. We are also more than happy to answer any further questions that may arise.

---

> > ### Comment · Reviewer_cVf8 · 2025-11-28
> >
> > Thank you for your response. I will retain my original evaluation.

---

### Official Review · Reviewer_Q6b1 · 2025-10-30

**Soundness:** 3
**Presentation:** 3
**Contribution:** 4
**Rating:** 6
**Confidence:** 3

**Summary:**

This paper presents OXtal, an all-atom diffusion model for molecular crystal structure prediction (CSP). The method directly learns the conditional joint distribution of intramolecular conformations and periodic crystal packing from input 2D chemical graphs. A key novelty is the introduction of the Stoichiometric Stochastic Shell Sampling ($S^4$) scheme, which avoids explicit lattice parameterization while capturing long-range interactions. Extensive experiments show that OXtal substantially improves over prior machine-learning CSP methods and even partially surpasses DFT-based methods on CCDC CSP blind tests.

**Strengths:**

1. To the best of the reviewer's knowledge, this is the first work to apply a full all-atom diffusion framework to molecular crystal structure prediction, which is a significant leap compared to previous approaches.

2. The empirical results are impressive, especially in realistic CCDC CSP blind tests, suggesting both effectiveness and efficiency of the proposed framework.

3. The $S^4$ representation is a novel idea for trading off explicit lattice parametrisation by sampling shells around stoichiometric units, which may offer a scalable solution for large-scale training of molecule CSP models.

**Weaknesses:**

While the proposed $S^4$ representation enables training on the proposed all-atom backbone model and the diffusion framework, there remain several issues and open questions that should be addressed or clarified, which are listed as follows:

1. The number of conformations inside each $S^4$ unit is not deterministic for one molecule crystal. In generation time, how many conformers one should generate per $S^4$ unit? Does this number affect downstream results?
2. Within each $S^4$ unit, the consistency of generated conformations is unclear. If conformations vary slightly or even significantly, how does one pick a representative conformer to compute RMSD or other evaluation metrics?
3. Although the selection of the lattice matrix is not unique, is it possible to recover one $S^4$ unit into a valid $(L,B)$ pair, i.e. an arbitrary unit cell? If so, can the detailed algorithm be provided?

**Questions:**

In addition to the above weaknesses specific to $S^4$, the reviewer has the following additional questions:

1. In the case where $n_S>1$, how are the evaluation metrics chosen? For instance, are they the average over all samples, or the best among them, or some other criterion?
2. In the CSP Blind Test 5 benchmark, the OXtal method outperforms prior DFT-based methods on most metrics, yet on the Sol_C subset there remains a noticeable gap. What are the suspected causes of the lower performance on Sol_C? Can the authors identify any corner-cases that the model struggles with and that would guide future improvements?

---

> ### Author Response · Authors · 2025-11-24
>
> We thank the reviewer for their positive review and constructive feedback on our paper. We appreciate that the reviewer has acknowledged our work as a “significant leap” forward compared to previous approaches. We are also thrilled to hear that the reviewer found our empirical results impressive and our $S^4$ lattice-free parameterization to be a scalable “novel idea.” We now address the key questions raised in the review grouped by theme.
>
> **Conformer Analysis**
>
> >The number of conformations inside each S4 unit is not deterministic for one molecule crystal. In generation time, how many conformers one should generate per S4 unit? Does this number affect downstream results?
>
> OXtal uses the same feature conformers for every molecular copy. We agree that some crystals contain multiple symmetry-inequivalent conformers (Z′ > 1), but it is a relatively rare occurrence: less than 10% of the crystals in our training dataset (and CSD) contain Z’ > 1. As detailed in the broader conformer analysis in our global response, the diffusion process always starts from random coordinates and the quality of the _feature conformer_ used for conditioning does not significantly impact the downstream results. Despite the same feature conformer, OXtal is still able to approximately solve some examples of Z’>1 (e.g. XXIII/XAFPAY02 in CSP Blind Test 6, where we obtain RMSD$_{15}$ of 1.914 Å).
>
> >Within each S4 unit, the consistency of generated conformations is unclear. If conformations vary slightly or even significantly, how does one pick a representative conformer to compute RMSD or other evaluation metrics?
>
> We use CSD’s standard pipeline COMPACK [1], which picks the closest conformer to compute RMSD$\_{1}$, and the closest cluster to compute RMSD$_{15}$.
>
> **Inference and Evaluation**
>
> >In the case where ns>1, how are the evaluation metrics chosen? For instance, are they the average over all samples, or the best among them, or some other criterion?
>
> When $n_S$ > 1, we report per-sample metrics ($Lat_S$ and $Rec_S$) as the average over all generated samples (we use $n_S = 30$ in our experiments). For the per-crystal metrics ($Lat_C$, $Rec_C$, and $\widetilde{\text{Sol}}_C$), we calculate metrics based on the best generated samples. This is consistent with DFT-based CSP literature which generates many examples per structure.
>
> >In the CSP Blind Test 5 benchmark, the OXtal method outperforms prior DFT-based methods on most metrics, yet on the Sol_C subset there remains a noticeable gap. What are the suspected causes of the lower performance on Sol_C?
>
> The performance gap is likely explained by the limited number of samples that we generate. Many of the DFT methods submit hundreds or thousands of samples, whereas we only evaluate 30, since we wish to highlight OXtal’s sample efficiency (see Figure 6). Here, we provide updated results from generating 500 samples for each target in CSP Blind Test 5, which matches the number of submitted structures by DFT methods. We see that with a larger sample size, we are in fact able to achieve the same approximate solve rate as DFT$_{\text{avg}}$. This result is detailed in Appendix F.2.
>
> | Model| $Col_S$ ↓| $Lat_S$ ↑| $Lat_C$ ↑| $Rec_S$ ↑| $Rec_C$ ↑| $\widetilde{Sol}_C$ ↑|
> |-|-|-|-|-|-|-|
> | DFT$_{\text{avg}}$ ($\overline{n_S} = 500$)| 0.003| 0.307| 0.556| 0.772| 0.681| **0.500**|
> | OXtal ($\overline{n_S} = 500$)| 0.009| 0.536| 1.000| 0.548| 1.000| **0.500**|

---

> ### Author Response · Authors · 2025-11-24
>
> **Unit Cell Extraction**
>
> We acknowledge the reviewer’s question regarding the feasibility of extracting a unit cell. We respond by first noting that for a point cloud of atoms specified by Cartesian coordinates, there are a few ways that have been detailed in the literature to identify non-unique unit cells, which can subsequently be reduced (e.g., with Niggli reduction) to unique representations. For example, the autocorrelation (“Patterson map”) of a periodic structure is a sum of interatomic vectors, and the lattice translations appear as the dominant, regularly spaced vectors [2]. A recent working paper [3] adapts such 3D Patterson-function analysis (a well-known approach in crystallography) as part of an automated algorithm for the initial identification of unit cells from a point cloud. Similar numerical methods have been used to find symmetries and structure in non-atomistic geometries [4]. Alternatively, methods include 3D fast Fourier transform, which is equivalent to 3D diffraction and can then be used to solve for the crystal structure [5]. We have now added a small note in the paper about this specific aspect.
>
> **Concluding Remarks**
>
> We thank the reviewer again for their time and effort in reviewing our paper. We believe our new rebuttal results and clarifications have answered all the great points raised by the reviewer. If the reviewer is satisfied with our responses, we would be encouraged if the reviewer would consider a fresher evaluation of our work with the rebuttal in context. We are also more than happy to answer any further questions that arise.
>
> **References:**
>
> [1] Chisholm, James Alexander, and Sam Motherwell. "COMPACK: a program for identifying crystal structure similarity using distances." Applied Crystallography 38.1 (2005): 228-231.
>
> [2] Terwilliger, T. C., S-H. Kim, and D. Eisenberg. "Generalized method of determining heavy-atom positions using the difference Patterson function." Foundations of Crystallography 43, no. 1 (1987): 1-5.
>
> [3] Kundu, Sumitava, Kaustav Chakraborty, and Avisek Das. "Algorithmic detection of the crystal structures from three-dimensional real-space analysis." arXiv e-prints (2024): arXiv-2407.
>
> [4] Pauly, Mark, et al. "Discovering structural regularity in 3D geometry." ACM SIGGRAPH 2008 papers. 2008. 1-11.
>
> [5] Zeng, Xiangyan, Bryant Gipson, Zi Yan Zheng, Ludovic Renault, and Henning Stahlberg. "Automatic lattice determination for two-dimensional crystal images." Journal of structural biology 160, no. 3 (2007): 353-361.

---

> > ### Comment · Reviewer_Q6b1 · 2025-11-27
> >
> > Thanks for the authors’ detailed response. The reviewer’s concerns have been fully addressed, and I have accordingly increased my rating and confidence. Just out of curiosity, is unit cell extraction already considered an efficiently solved problem?

---

> ### Author Response · Authors · 2025-11-28
>
> We sincerely thank the reviewer for all their feedback and for raising their score to an 8 and increasing their confidence to a 4.
>
> Regarding their question on unit cell extraction, we believe unit cell extraction is essentially a solved problem in reciprocal space (e.g. diffraction data or FFTs from periodic point clouds) and also in direct space for clean lattice point sets. Nevertheless, in certain scenarios (e.g. substantial amount of noise or lattice distortion), the problem is yet to be efficiently resolved. If the reviewer has any additional questions, please do not hesitate to let us know.

---

### Official Review · Reviewer_5p8A · 2025-11-01

**Soundness:** 2
**Presentation:** 4
**Contribution:** 3
**Rating:** 6
**Confidence:** 3

**Summary:**

This paper presents OXTAL, a large-scale diffusion model for organic crystal structure prediction (CSP). The model uniquely abandons explicit equivariant architectures, instead training a non-equivariant transformer on a massive 600K-structure dataset. It introduces "Stoichiometric Stochastic Shell Sampling" (S⁴) to train on local, non-periodic atomic blocks, aiming to learn both intramolecular conformation and intermolecular packing. The method reports state-of-the-art results among machine learning baselines and achieves performance competitive with DFT methods at orders-of-magnitude lower inference cost.

**Strengths:**

1. Performance and Efficiency: The model demonstrates impressive results, significantly outperforming existing ML baselines and achieving a massive reduction in computational cost compared to traditional DFT-based CSP workflows.

2. Scalability and Data: The work successfully scales a generative model to an exceptionally large and diverse dataset (600K CSD structures), a non-trivial data engineering and modeling achievement.

3. Clarity and Presentation: The figures and tables are clear, informative, and visually compelling.

**Weaknesses:**

1. Input Conformer Dependency: The model relies on a pre-optimized GFN2-xTB 3D conformer as input. This introduces a significant confounding variable and undermines the claim of learning a "joint distribution" ab initio; it appears to be refining a pre-calculated conformer rather than generating one from scratch.

2. Weak Theoretical Justification (Prop. 1): Proposition 1 merely describes a standard geometric surface-to-volume scaling law $O(T^{-1/3})$. It provides no evidence that the model learns true long-range periodicity or order from the local S⁴ crops.

3. Lack of Ablation Studies: The paper is missing crucial ablation studies. The individual contributions of the S⁴ sampling scheme, the non-equivariant architecture, and the large dataset are not decoupled, making it difficult to assess the true sources of the performance gain.

**Questions:**

1. Learned Representations (Fig. 4): In Figure 4, the generated molecules appear to have an identical orientation, suggesting the model may have only learned a packing arrangement rather than diverse, atomic-resolution geometries. Is this a common failure mode, and what does it imply about the model's learned representations?

2. Robustness to Input: How robust is OXTAL's performance to the quality of the initial GFN2-xTB conformer? What happens if the input is a high-energy or physically unrealistic conformer?

3. Choice of Architecture: Given that the S⁴ method trains on local atomic blocks, why was an explicitly SE(3)-equivariant denoiser (e.g., based on Equiformer or E(3)-GNNs) not used?

---

> ### Author Response · Authors · 2025-11-24
>
> We thank the reviewer for their review and constructive feedback on our paper. We are pleased that the reviewer has acknowledged our impressive performance and efficiency results and has found our work on scaling OXtal to be a “non-trivial data engineering and modelling achievement.” We now address the key questions raised in the review grouped by theme.
>
> **Conformer Analysis**
>
> We agree that it is important to understand the role of the conformer used for the feature embedding. As mentioned in our global response and detailed in Appendix E, the RDKit ETKDG + GFN2-xTB conformer is used solely as a feature-conditioning signal for the molecular encoder, providing a computational prior from quantum-chemical approximations. We therefore view OXtal as computationally ab initio conditioned on this prior, by not using empirical structural information from the target crystal.
>
> The diffusion process always starts from random atomic coordinates, and Figure 13 in Appendix E shows that the conformer used for conditioning is indeed different from the true crystal conformer to be predicted (panel a). Furthermore, the structures that OXtal generate via the diffusion processes differ from the conformer used for feature conditioning (panel b). These results remain true for feature conformers generated by different methods, and are especially apparent for flexible molecules.
>
> >Robustness to Input: How robust is OXTAL's performance to the quality of the initial GFN2-xTB conformer? What happens if the input is a high-energy or physically unrealistic conformer?
>
> We see from Figure 14 in Appendix E that despite using different conformers for feature conditioning (e.g. RDKit, UFF, MMFF), the crystal packing prediction performance of OXtal remains relatively unchanged. When bond information is corrupted significantly, the model performance falls drastically since it fundamentally becomes a different molecule. However, when only the atom coordinate features are perturbed, the model appears robust to different conformer priors.
>
> **Inference Analysis**
>
> >Learned Representations (Fig. 4): In Figure 4, the generated molecules appear to have an identical orientation, suggesting the model may have only learned a packing arrangement rather than diverse, atomic-resolution geometries.
>
> We provide a qualitative visualization of some example packings generated by OXtal in Figure 19 in Appendix H.1. Although OXtal may favor planar packings in the same orientation, which are more prevalent in the training dataset, many samples do present a more complex herringbone packing. In Figure 19, this is evident in the XATJOT co-crystal, which does not collapse the two different molecular components into the same orientation. We have now revised the main text to note the potential limitations.
>
> >Weak Theoretical Justification (Prop. 1): Proposition 1 merely describes a standard geometric surface-to-volume scaling law O(T^-⅓). It provides no evidence that the model learns true long-range periodicity or order from the local S⁴ crops.
>
> We agree with the reviewer that Prop. 1 does not directly speak to the long-range periodicity that can be learned via $S^4$ crops. However, we have provided an empirical analysis of how OXtal performs on larger inference blocks in Figure 15 in Appendix F.1. Even though OXtal was trained with a maximum token budget of 640, it is able to preserve lattice periodicity for up to thousands of tokens at inference time. Furthermore, conformer generation remains stable as well for these larger blocks. This analysis supports the long-range generalizability of OXtal to generate larger periodic packings, and we provide a visualization of the approximately solved larger packing for ANTCEN in Figure 16, which contains over 2,400 tokens.
>
> **Ablation Studies**
>
> We thank the reviewer for their suggestion and have provided additional ablation studies in our global response. In summary, the performance gain is a combination of both the $S^4$ cropping method and the model scaling derived from a non-equivariant architecture.
>
> >Choice of Architecture: Given that the S⁴ method trains on local atomic blocks, why was an explicitly SE(3)-equivariant denoiser (e.g., based on Equiformer or E(3)-GNNs) not used?
>
> The main bottleneck of explicitly equivariant architectures is the scalability factor. As we see from our equivariance ablation with EquiformerV2, a non-equivariant model is essential to being able to scale OXTal to the size of larger molecular crystals without sacrificing model capacity.
>
> **Concluding Remarks**
>
> We thank the reviewer again for their time and effort in reviewing our paper. We believe our new rebuttal results and clarifications have answered all the great points raised by the reviewer. If the reviewer is satisfied with our responses, we would be encouraged if the reviewer would consider a fresher evaluation of our work with the rebuttal in context. We are also more than happy to answer any further questions that arise.

---

> ### Author Response · Authors · 2025-11-28
>
> We kindly remind the reviewer that the discussion period will be ending soon. Therefore, we would greatly appreciate it if the reviewer could share any outstanding questions they may have.

---

### Official Review · Reviewer_zXpA · 2025-11-01

**Soundness:** 3
**Presentation:** 3
**Contribution:** 3
**Rating:** 6
**Confidence:** 3

**Summary:**

This paper introduces OXTAL, a large-scale, all-atom diffusion model designed to tackle the problem of organic crystal structure prediction (CSP). The model aims to predict the complete 3D crystal structure—encompassing both the molecule's internal conformation and its periodic packing arrangement—conditioned only on the 2D molecular graphs.

**Strengths:**

1. The idea of the lattice-free approach is interesting. Instead of "generate a valid unit cell," it becomes "generate a locally consistent, periodic atomic environment". This is a fundamental and elegant simplification that is physically motivated by the local-to-global nature of crystallization.

2. The decision to abandon explicit equivariance in favor of a massive model, massive data, and simple data augmentation is a bold and counterintuitive claim compared to most current research literature in geometric ML for this application.

3. The cost-performance plot in Figure 7 is an amazing result. It demonstrates that the proposed OXTAL provides a solution that is orders of magnitude cheaper than the baseline models.

**Weaknesses:**

1. The paper's most important claim is that abandoning explicit equivariance is an interesting idea. This is a central, non-obvious claim. However, there is no ablation study to back this up. The ablation study regarding other components in the $S^4$ is also missing.


2. Some baseline models are missing in the experiments, including MatterGen[1] and ADiT[2]. It is important to compare the proposed method with the latest methods in this evolving field.


3. Algorithm 1 introduces several new hyperparameters, including the shell radius, which is particularly critical. The paper provides no sensitivity analysis for these choices.



[1] Claudio Zeni, Robert Pinsler, Daniel Zugner, Andrew Fowler, Matthew Horton, Xiang Fu, Zilong Wang, Aliaksandra Shysheya, Jonathan Crabbe, Shoko Ueda, et al. A generative model for inorganic materials design. Nature, 639(8055):624–632, 2025.


[2] Chaitanya K. Joshi and Xiang Fu and Yi-Lun Liao and Vahe Gharakhanyan and Benjamin Kurt Miller and Anuroop Sriram and Zachary W. Ulissi. All-atom Diffusion Transformers: Unified generative modelling of molecules and materials. International Conference on Machine Learning, 2025.

**Questions:**

1. Could you provide an ablation study comparing OXTAL's non-equivariant Pairformer trunk against a state-of-the-art equivariant trunk (e.g., an equivariant transformer or GNN) trained with the same $S^4$ cropping method?


2. Can you provide a baseline comparison showing the benefit of the $S^4$ "lattice-free" scheme?

3. How sensitive is the model's performance to the choice of the shell radius?

4. Could you please elaborate on how the model handles co-crystals and solvates with specific stoichiometries? How is the model conditioned at inference time to produce, given that the input is just the SMILES strings?

---

> ### Author Response · Authors · 2025-11-24
>
> We thank the reviewer for their positive review and constructive feedback on our paper. We appreciate that the reviewer has acknowledged our innovative lattice-free non-equivariant approach and found our decision to abandon equivariance in favor of scaling to be a “bold” choice that goes against the grain of existing literature. We are also thrilled to hear that the reviewer found our cost-performance plot in Fig. 7 to be an “amazing result.” We now address the key questions raised in the review grouped by theme.
>
> **Ablations on Equivariance and $S^4$**
>
> We agree with the reviewer’s feedback that our central claim of using a non-equivariant model could be bolstered through further ablation. As such, we have now included an equivariance ablation study with EquiformerV2 in our global response. These results show that using a non-equivariant model is essential to being able to scale OXtal to the size of larger molecular crystals. This ablation strengthens our claim that abandoning equivariance is a frictionless way of scaling.
>
> >Can you provide a baseline comparison showing the benefit of the $S^4$ "lattice-free" scheme?
>
> We thank the reviewer for this insightful suggestion and present ablations that highlight the benefit of using $S^4$ cropping in our global response. Overall, it is clear that our crystallization-inspired $S^4$ approach outperforms other standardized cropping methods.
>
> >How sensitive is the model's performance to the choice of the shell radius?
>
> We thank the reviewer for their suggestion and provide results for different choices of $S^4$ radius size in our global response. We see that although there is some minimal change in performance between different radius sizes, all three settings for $S^4$ still outperform other cropping methods, suggesting the model is robust to reasonable choices of shell radius size.
>
> **Additional Baselines**
>
> We acknowledge the reviewers' request for additional baselines through their quote:
>
> >Some baseline models are missing in the experiments, including MatterGen[1] and ADiT[2].
>
> We would like to politely clarify that both MatterGen and ADiT fundamentally solve a different problem of _de novo_ inorganic crystal generation, rather than molecular crystal structure prediction (CSP). Specifically, OXtal is designed to predict the 3D crystal packing conditioned on a specific 2D molecular structure. In contrast, MatterGen and ADiT are not conditioned on molecular structure information, but rather generate the atom types, in addition to unit cell lattice, and atom positions for the “new” crystals. Furthermore, MatterGen and ADiT are designed for inorganic crystals, which are usually smaller, do not contain molecular conformers, and are characterized by strong covalent or ionic bonds (as opposed to the weak, longer-range bonds in molecular crystals). Both of these models would therefore require significant modifications and retraining to adapt to our use case of CSP and would constitute a separate research endeavor altogether.
>
> We believe the closest generative model for inorganic crystals is actually FlowMM [1], which does generate packings for inorganic crystals conditioned on atomic and lattice information. We have thus performed a zero-shot evaluation of FlowMM as an additional baseline here on the Rigid and Flexible datasets:
>
> | $\text{Dataset}$  | $Col_s$ ↓    | $Lat_S$ ↑    | $Lat_C$ ↑      | $Rec_S$ ↑      | $Rec_C$ ↑      | $\widetilde{Sol}_C$ ↑     |
> |--------------|-------|-------|-------|-------|-------|------------|
> | Rigid    | 0.518 | 0.016 | 0.090 | 0 | 0 | 0      |
> | Flexible  | 0.772 | 0 | 0 | 0 | 0 | 0      |
>
> As expected, FlowMM performs extremely poorly since it is designed for inorganic crystals, which form a vastly different data distribution.
>
> **Handling Specific Stoichiometries**
>
> For co-crystals and solvates with specific stoichiometries, we specify the specific stoichiometric ratios of the individual components at inference time. For example, if a co-crystal has a 2:1 stoichiometry, we provide the SMILES strings for both components, and tell the model to generate 2 copies of the first component for every 1 copy of the second component.
>
> **Concluding Remarks**
>
> We thank the reviewer again for their time and effort in reviewing our paper. We believe our new rebuttal results and clarifications have answered all the great points raised by the reviewer. If the reviewer is satisfied with our responses, we would be encouraged if the reviewer would consider a fresher evaluation of our work with the rebuttal in context. We are also more than happy to answer any further questions that arise.
>
> **References:**
>
> [1] Miller, Benjamin Kurt, et al. "Flowmm: Generating materials with riemannian flow matching." arXiv preprint arXiv:2406.04713 (2024).

---

> > ### Comment · Reviewer_zXpA · 2025-11-26
> >
> > Thanks for the response. I keep my assessment.

---

> > > ### Author Response · Authors · 2025-11-27
> > >
> > > Thank you again for your thoughtful review and for taking the time to read and engage with our rebuttal. We wanted to briefly check whether the additional experiments and clarifications we provided address your main concerns regarding (i) the role of equivariance vs. data augmentation, (ii) the choice and sensitivity of the Algorithm 1 hyperparameter, and (iii) the baseline ablations that disentangle the contributions of our components. We would be happy to answer any lingering questions on these points or other open concerns while the discussion period remains open.
> > >
> > > If the reviewer feels satisfied with our rebuttal responses, we would also love an opportunity to engage further on any aspect that the reviewer feels could help enable them to more strongly endorse our paper and warrant a potential score upgrade. Please do let us know!

---

### Author Response · Authors · 2025-11-24
**Global Response**

We thank the reviewers for their time, energy, and constructive comments that have allowed us to strengthen our paper with new ablations and analyses. We are particularly heartened to hear that reviewers appreciated our novel lattice-free diffusion formulation unlocked through $S^4$ sampling, describing it as an “elegant simplification that is physically grounded” while being an innovative and “scalable solution for large-scale training” (zXpA, 5p8A, Q6b1, cVf8). Furthermore, we are thrilled that reviewers (O6b1, cvf8) recognized that OXtal presents one of the first comprehensive all-atom diffusion frameworks for molecular CSP enabled by large-scale training on 600K CSD structures (5p8A, cVf8). We are also grateful that the reviewers found our empirical results to be “impressive, especially in realistic CCDC CSP blind tests” (Q6b1), with OXtal delivering DFT-comparable accuracy at orders-of-magnitude lower computational cost (zXpA, 5p8A, cVf8).

Here, we address shared clarification points across reviewers and summarize additional experimental results for S4 ablations, equivariance ablations, and conformer analysis that are included in our revised pdf.

**Performance gain from S4 (Reviewer zXpA, 5p8A):** We isolate the performance gain from our novel $S^4$ cropping scheme by comparing it to standard KNN and Centroid Radius methods. All models are trained for 50k steps, with results averaged across all 5 datasets (see Appendix D.1 for detailed results).

|$\text{Crop}$| $Col_s$ ↓| $Lat_S$ ↑| $Lat_C$ ↑| $Rec_S$ ↑| $Rec_C$ ↑| $\widetilde{Sol}_C$ ↑|
|-|-|-|-|-|-|-|
| KNN | 0.2390 | 0.3566 | 0.7650 | 0.1940 | 0.4000 | 0.0840 |
| Centroid Radius | 0.1548 | 0.3978 | **0.8502** | 0.2690 | 0.5596 | 0.0930 |
| $S^4$ | **0.0816** | **0.4390** | 0.8296 | **0.2736** | **0.7356** | **0.1336** |

$S^4$ performs the best for all but one metric, since it does not suffer from the same pitfalls of KNN and Centroid Radius cropping (see Figure 9 in Appendix B).

**Effect of S4 shell radius size (Reviewer zXpA, 5p8A, cVf8):** We investigate the model’s sensitivity to the choice of $S^4$ shell radius size. All models are trained for 50k steps, with results averaged across all 5 datasets (see Appendix D.2 for detailed results).

| $r_{cut}$ | $Col_s$ ↓| $Lat_S$ ↑ | $Lat_C$ ↑ | $Rec_S$ ↑ | $Rec_C$ ↑ | $\widetilde{Sol}_C$ ↑ |
|-|-|-|-|-|-|-|
| 3.5    | 0.0918 | 0.4348 | **0.8502** | 0.2708 | 0.6456 | **0.1940**|
| 4.5  | **0.0816** | **0.4390** | 0.8296 | **0.2736**   | **0.7356**   | 0.1336 |
| 5.5    | 0.1442 | 0.4230 | 0.8214 | 0.2762 | 0.5828 | 0.1216 |

We see that there is minimal difference in performance, suggesting that the model is robust to reasonable choices in $S^4$ radius size. Specifically, all three radius settings for $S^4$ outperform the other cropping methods as reported above.

**Equivariance ablation (Reviewer zXpA, 5p8A, cVf8):** We evaluate the performance of EquiformerV2 trained with the same $S^4$ cropping scheme and training pipeline. Due to memory constraints, we are forced to reduce the model size of EquiformerV2 to only 15M total parameters, as opposed to the 100M used for OXtal. Results are averaged across all 5 datasets (see Appendix D.3 for detailed results).

| $\text{Model}$ | $Col_s$ ↓ | $Lat_S$ ↑ | $Lat_C$ ↑ | $Rec_S$ ↑ | $Rec_C$ ↑ | $\widetilde{Sol}_C$ ↑  |
|-|-|-|-|-|-|-|
| EquiformerV2 | 0.1164 | 0.0018 | 0.0544 | 0.0018 | 0.0120 | 0 |
| OXtal | **0.0816** | **0.4390** | **0.8296** | **0.2736** | **0.7356** | **0.1336** |

These results show the necessity of using a scalable, non-equivariant model: enforcing equivariance forces us to drastically limit our parameter size, which is insufficient for modelling the complex crystallization landscape. This critical bottleneck validates our design choice of scaling via a non-equivariant model and breaking free from lattice constraints while softly enforcing symmetry constraints.

**Conformer analysis (Reviewer 5p8A, Q6b1):** We first clarify that the RDKit ETKDG + GFN2-xTB conformer is used solely as a conditioning feature for the molecular encoder; the generative diffusion process always starts from random atomic coordinates, so OXTAL remains fully ab initio and also does not use oracle structural information from the target crystal. We have updated Appendix E to quantify the influence of this initialization:
1. **Influence of the initial conformer relative to ground truth:** We show that the GFN2-xTB conformers often differ substantially from the crystal conformers.
2. **Distance of the feature conformer relative to the prediction:** OXTAL does not merely reproduce the conditioning conformer; the learned denoiser generates final conformations that systematically differ from the feature conformer.
3. **Robustness to initial conformer quality:** We evaluate robustness by replacing GFN2-xTB with alternative or perturbed conformers (RDKit-ETKDG, UFF, MMFF94, noisy and ground-truth conformers), observing no significant change in generation performance.

---

### Author Response · Authors · 2025-12-03
**Summary for AC**

We sincerely thank the AC for their extra time and effort in these challenging circumstances. Below, we provide a summary of our work and the initial feedback received.

**Summary of our work**

We present OXtal, the first large-scale all-atom diffusion model for molecular crystal structure prediction (CSP). This problem is such a landmark challenge in computational chemistry that in 1988, the editor of Nature, John Maddox, famously declared, “One of the continuing scandals in the physical sciences is that it remains in general impossible to predict the structure of even the simplest crystalline solids from a knowledge of their chemical composition.”

By abandoning explicit equivariance and unit cell representations in favor of scaling and our novel Stoichiometric Stochastic Shell Sampling ($S^4$) procedure, our work represents a step change in progress for a problem that has eluded ML efforts thus far. Initial reviews have championed our work for:

- An "innovative" and “elegant simplification that is physically grounded,” producing a “scalable solution for large-scale training” that is “bold and counterintuitive” (`cVf8`, `5p8A`, `Q6b1`, `zXpA`).
- “Amazing” and “impressive” empirical results especially in delivering DFT-comparable accuracy at orders-of-magnitude lower computational cost for realistic CCDC Blind Test crystals (`zXpA`, `5p8A`, `Q6b1`, `cVf8`).
- “Among the most technically compelling works to date on molecular CSP” enabled by "massive model and massive data," which is “a non-trivial data engineering and modeling achievement” and "is a significant leap compared to previous approaches" (`cVf8`, `zXpA`, `5p8A`, `Q6b1`).

**Overview of initial discussions**

Reviewers highlighted the following points of improvement, all of which we have thoroughly addressed:

- We present several new ablation studies highlighting the impact of equivariance, $S^4$ cropping, shell radius side, and model size.
- We provide an in-depth conformer analysis detailing the differences between the initial conformer and crystal conformer, the ability of OXtal to generate new conformations, and the robustness of OXtal to initial conformer quality.
- We evaluate an additional baseline from inorganic crystal generation and show how it is ineffective for this use case, which further emphasizes the impact of our contribution in this space.
- We conduct additional inference analyses that show the long-range generalizability of $S^4$, OXtal’s ability to match average DFT performance when allowed the same number of generated samples, and a qualitative investigation into the diversity of generated samples.

This work marks a significant shift in the way that ML can help address this problem of molecular CSP, and we hope it marks the beginning of a new chapter in molecular materials design.

---

### Meta-Review · Area_Chair_StGE · 2026-01-05

**Summary:**

In this submission, the authors develop a large-scale all-atom diffusion model for organic crystal structure prediction, achieving strong performance on this challenging task compared to other ML-based solutions. Technically, the proposed method achieves high efficiency and scalability by 1) abandoning explicit equivariance constraints in the model architecture and 2) leveraging Stoichiometric Stochastic Shell Sampling (S$^4$) to capture long-range interactions among atoms. Experiments demonstrate the potential and rationality of the proposed method to some degree.

**Reviewer Concerns:**

The main concerns of the reviewers include

1. The rationality and the usefulness of S$^4$.

2. The robustness of the proposed method to the quality of initial conformation and hyperparameter settings.

3. The generalizability of the proposed method.

4. The solidity of the comparison experiments.

In the rebuttal phase, the authors made efforts to address the above concerns, including adding ablation studies, more baselines, and more robust hyperparameter tests.

**Reviewer Scores:**

Because most concerns have been resolved, in my opinion, and I think the reviewers would have maintained or increased their scores.

---

### Decision · Program_Chairs · 2026-01-26

Accept (Poster)